# Analysis of UTCI index during heat waves in Serbia

Milica M. Pecelj[123], Milica Z. Lukić[4], Dejan J. Filipović[4], Branko M. Protić[4], Uroš M. Bogdanović[5]

[1]Geographical Institute Jovan Cvijić Serbian Academy of Science and Arts, Belgrade, Serbia
[2] Department of Geography, University of East Sarajevo, East Sarajevo, RS, Bosnia and Herzegovina
[3]South Ural State University, Institute of Sports, Tourism and Service, Chelyabinsk, Russia
[4]University of Belgrade – Faculty of Geography, Belgrade, Serbia
[5]University of Belgrade, Faculty of Organisational Sciences, Belgrade, Serbia

*Correspondence to*: Milica M. Pecelj (milicapecelj@gmail.com)

**Abstract.** The objective of this paper is to assess the bioclimatic conditions in Serbia during summer in order to identify
biothermal heat hazard. Special emphasis is placed on the bioclimatic index UTCI (Universal Thermal Climate Index), whose purpose is to evaluate the degree of thermal stress that the human body is exposed to. For this research, mean daily and hourly (7 and 14 CET) meteorological data from three weather stations (Mt. Zlatibor, Novi Sad, Niš) have been collected for the period from 1998 to 2017. In order to identify patterns of biothermal heat stress conditions, the thresholds of daily UTCI index (UTCI $\geq 38°C$ referred to as very strong heat stress-VSHS) at 14 CET, are compared with the thresholds of daily maximum of
air temperature ($t_{max} \geq 35°C$ referred to as hot day-HD) which are further termed as heat wave events (VSHSE and HWE). The findings show that the UTCI heat stress category "very strong heat stress" at 14 CET indicates heat waves. The most extreme heat wave events occurred in 2007, 2012, 2015 and 2017. Moreover, three heat wave events (HWE) in Niš that occurred in July 2007 lasted 3, 10 and 4 days. Heat wave events (HWE) and very strong heat stress events (VSHSE) recorded in July 2007 (lasted 10 days each), 2012 (lasted 9 and 12 days) 2015 (lasted 7 and 10 days) were of the longest duration and are considered
to be the indicators of biothermal heat hazard. The daily $UTCI_{14h}$ heat stress becomes more extreme in terms of severity and heat wave duration up to very strong heat stress (VSHS).
**Key words**: Bioclimatic conditions, Universal Thermal Climate Index (UTCI), heat waves, VSHS.

## 1. Introduction

Extreme weather events such as heat waves, floods, droughts or storms have shown an increased frequency in recent decades
(Brown et al., 2008; Easterling et al., 1997; IPCC, 2012; Rahmstorf and Coumou, 2011; Seneviratne and Nicholls, 2012). Not only are heat and drought events of great importance in most Mediterranean climate regions, but also in most of southern and south-eastern Europe, because of the diverse and costly impact that they have on various economic sectors and on the environment (Peña-Gallardo et al., 2019). In their study Spinoni et al. (2015) made a list of the most severe drought events which occurred in Europe from 1950 to 2012. They singled out the Balkans (particularly Serbia) as an area that is susceptible
to extreme weather and drought (the longest drought was registered in 2007, and the most severe drought event in 2011).

Furthermore, in 2012 the Balkan Peninsula and the south-eastern Europe experienced the hottest summer and one of the worst droughts in nearly 40 years (Unkašević and Tošić, 2015).

Extremely high temperatures, especially during the summer months, beg the question as to how heat could affect everyday life of humans. Heat waves as a hazard often have a negative effect, causing heat stress in the human body. They have both direct effects on human health, affecting the body's physiological responses and functions and indirect effects on human health, increasing challenges to food and water safety (Lee et al., 2019). Bioclimatic condition provides a base for considering the effects of climatic conditions on humans and gives special importance to the social factors that mitigate or reinforce the consequences of environmental changes (Bleta et al., 2014). The impact of the weather and climate on humans is most commonly described as the biothermal condition of a certain area, presented by bioclimatic indices. Over the last century many models and indices have been developed for the assessment of human exposure to heat, ranging from simple physical instruments designed to imitate the human heat exchange with the environment, to complex thermophysiological models that simulate external and internal body heat transfer and allow detailed simulation of different work load, clothing, and climate scenarios (Havenith and Fiala, 2015). According to Epstein et al. (2006), there are over 100 heat stress indices that could describe extreme bioclimatic conditions in relation to humans. Many thermal indices have been developed for the purpose of describing the complex conditions of heat exchange between the human body and its thermal environment (Fanger, 1970; Landsberg, 1972; Parsons, 2003). Those are two meteorological parameter indices – the ones used for cold environment conditions combining the air temperature and the wind velocity (Osczevski and Bluestein, 2005; Siple and Passel, 1945;), and the ones used for heat environment conditions combining the air temperature and humidity (Masterton and Richardson, 1979; Steadman, 1984; Yaglou and Minard, 1957). Considering their shortcomings, i.e. the fact that they are not universally applicable to all climatic regions, including different seasons, the dominance in the analysis of biothermal conditions and thermal stress over the last 30 years has been taken over by so-called heat budget indices (Blazejczyk, 1994; Hoppe, 1999; Jendritzky, 2012). The heat budget indices are based on the thermal exchange between humans and the environment. Methodologically, they use variables related to meteorological (air temperature, wind velocity, radiation, air humidity) and physiological processes (most commonly metabolic heat) and clothing insulation. The issue that should be emphasized during the analysis of the thermal stress is the influence that extreme temperatures have on the physiological parameters in humans. According to McGregor and Vanos (2018), generated heat load can undermine the human body's ability to hold its core temperature within the range of optimal physiological achievement.

For the purpose of estimating the thermal effect of the environment on the human body, the total effects of all the thermal components are considered by the UTCI index. The UTCI is heat budget index considering both physiological and meteorological parameters describing the physiological heat stress that the human body experiences while achieving thermal equilibrium with the surrounding outdoor environment (Błażejczyk et al., 2013). Compared to other indices, UTCI is more sensitive to changes in all of the environment parameters, particularly air temperature, solar radiation, humidity, and wind speed (Błażejczyk et al., 2012). According to Jendritzky et al. (2012) the UTCI evaluates the outdoor thermal environment for biometeorological applications by simulating the dynamic physiological response with a model of human thermoregulation

together with a modern clothing model. UTCI provides a multiple opportunity for investigation and has a wide application in weather forecast for outdoor activities, appropriate behavior and climate therapies, and extreme thermal stress alerts. These analyses can have application in tourism, health sector, and urban and regional planning (Jendritzky et al., 2012). In the past decade, a large number of publications in Europe concern analyses of bioclimatic condition in accordance with the UTCI index (Błażejczyk et al., 2014; Błażejczyk et al., 2017; Bleta et al., 2014; Di Napoli et al., 2018; Di Napoli et al., 2019; Matzarakis et al., 2014; Milewski, 2013; Nastos and Matzarakis 2012; Nemeth, 2011; Tomczyk and Owczarek, 2019; Urban and Kyselý, 2014).

Although over recent years in Serbia articles have been published containing analysis of bioclimatic condition conducted by means of other heat budget indices (Basarin et al., 2016; Basarin et al., 2018; Lukić et al., 2019; Milovanović et al., 2017; Pecelj et al., 2017; Pecelj et al., 2018; Stojicević et al., 2016), detailed bioclimatic condition has not yet been thoroughly analysed by means of the UTCI index, especially in the case of heat waves. The heat wave is a prolonged period of unusually high air temperature that causes changes in everyday life and may cause health and wellbeing effects of heat stress. Even though a heat wave is understood as a meteorological event, its significance and influence could not be adequately presented without showing its clear impact on man (Robinson, 2001). There is a need to study the evolution of such indices regarding heat stress that affects the human thermoregulatory system as a result of heat exchange between the body and its thermal environment.

Considering extreme temperature is one of the most significant climatic parameters in the universal context of climate change, analyses of heat waves in Serbia have been performed in several different approaches and all of them show a growing trend in max temperature (Drljača et al., 2009; Unkašević and Tošić, 2009a; Unkašević and Tošić, 2009b; Unkašević and Tošić, 2013) and heat waves (Unkašević and Tošić, 2011; Malinović-Milićević et al., 2013; Unkašević and Tošić, 2015, Milićević-Malinović et al., 2016) especially since 2000.

Keeping this fact in mind, the basic idea of this study is to conduct the analysis of human bioclimatic conditions in Serbia in the last 20 years, where according to the previous studies mentioned above, warming has been perceived and recorded, especially since 2000. Determining extreme weather event from the aspect of heat budget indices allows other meteorological parameters to be taken into account because high temperatures and humidity generate a heat load more rapidly, unless the wind and direct radiation are taken into account. The analysis of bioclimatic conditions implies the determination of UTCI thermal stress during the summer months and different heat stress that has occurred in the past 20 years in different geographical landscapes in Serbia. In addition to UTCI thermal stress, extreme temperature thresholds were analysed in order to observe the biothermal stress related to the heat wave. The conducted human bioclimatic evaluation of UTCI thermal stress in Serbia was considered to be of great importance due to the identification of biothermal heat hazard and the study of the evolution of such indices regarding climate change. This evaluation is aimed at addressing the following topics in this study:

- providing a comprehensive assessment of the human heat stress associated with UTCI index and
- comparing it to heat waves defined by meteorological parameters.

*Study area*

The Republic of Serbia is located in south-eastern Europe, in the area of the southern Pannonian Plain and the central Balkans. Northern Serbia is mainly flat, while its central and southern areas consist of highlands and mountains (Gocić and Trajković, 2014), as the map of the relief characteristics shows (Figure 1). The territory of Serbia is characterised by temperate continental and mountain climate and the spatial distribution of climate parameters is determined by geographical location. According to

105 Koppen-Geiger Climate Classification (Kottek et al., 2006), the territory of Serbia is characterised by warm temperate humid climate type with warm summers, Cfb type with maximum precipitation during late spring and early summer. The research involved studying three synoptic stations located in different geographical areas of Serbia: (i) Novi Sad, (ii) Niš and (iii) Mt. Zlatibor (Figure 1).

**Figure 1**. Relief map of Serbia with the studied areas: 1-Novi Sad, 2-Niš, 3-Zlatibor (left). Map of the average number of tropical days in Serbia per year for the period 1981-2010 (in the middle) (source: Republic Hydro-meteorological Service of Serbia, RHMSS) and a map of geographical location of Serbia in Europe (right) [Maps were created using QGIS 3.8 software based on the official European Commission's (Eurostat) datasets, available at https://ec.europa.eu/eurostat/web/gisco/geodata/reference-data/, (Map ratio: 1:1900000; Map projection: WGS 84/UTM, Zone

34N, the official national coordinate system)].

The first weather station (at an altitude of 86 m) is located in the territory of Novi Sad, which is the administrative urban center of Vojvodina province and South Bačka District (Figure $1_1$). The city is located in the southern part of the Pannonian Basin, on the Danube River bank near Mt. Fruška Gora, and the national park bearing the same name. Novi Sad has a temperate

continental climate, summers are warm and winters are cold with a small amount of snow (Lazić et al., 2006). The second station is Niš (at an altitude of 202 m) and it is located in the Niš Fortress. This city is the administrative urban center of the Nišava District in southern Serbia and it is situated in the Nišava valley, located in the central part of the spacious geological depression called the Nišava Basin (Figure $1_2$). According to Köppen's climate classification, the Nišava valley belongs to the Cfwax type – the Danube type of moderately warm and humid climate characterized by hot summers (the highest precipitation

is recorded at the beginning of the summer) and somewhat dry winters (Prokić, 2018). Compared to other cities in this valley (Dimitrovgrad, Pirot and Bela Palanka), Niš is the hottest one with an average annual temperature of 11.8°C (Prokić, 2018).

The third station is Zlatibor, a mountain in western Serbia, which belongs to the mountain range of the Dinaric Alps (Figure $1_3$). Zlatibor weather station lies at an altitude of 1029 m a.s.l. In the area of Zlatibor plateau mountains meet air currents from the Adriatic Sea, which, as it can be assumed, creates a favorable climate and for this reason Zlatibor has already been

confirmed as a climatic resort (Pecelj et al., 2017). The mountain climate of Zlatibor involves long and cold winters, short and fresh summers, and less pronounced spring and autumn.

## 2. Materials and methods

The present study implements the methodological approach of Universal Thermal Climate Index (UTCI). As a thermal comfort indicator, the UTCI considers combined meteorological and physiological parameters describing thermal comfort through the evaluation of human energy balance. In terms of physiological conditions, the metabolic rate plays an important role. Metabolic processes in the human body create heat that is constantly exchanged with the environment achieving a state of thermal equilibrium in the body for maintaining a constant body temperature. The amount of the heat that is produced and released depends on the following: physical activity, clothing, sex, age, body mass, diet, mental state, health, external conditions, acclimatization, etc. The unit of „MET" was introduced as a measure of physical activity. 1 MET equals the heat release of 58.2 $W.m^{-2}$ from the average skin surface area of an adult (1.8 $m^2$). It is equal to the energy rate produced per unit surface area of an average person seated at rest (ANSI/ASHRAE Standard 55). According to ISO 8996, the metabolic heat energy of a person moving at the speed of $1.1 m.s^{-1}$ is M = 135 $W.m^{-2}$, i.e. 2.3 MET.

*The Universal Thermal Climate Index (UTCI)*

For valid assessment of the outdoor thermal environment in the fields of public weather services, public health systems, urban planning, tourism and recreation and climate impact research, the Universal Thermal Climate Index (UTCI) has been developed based upon the most recent scientific progress in human thermo-physiology in biophysics and heat exchange theory. The UTCI is the result of an approach which was developed in the International Society of Biometeorology (ISB) Commission 6 and was later improved by COST Action 730 (Jendritzky et al., 2012). The design of UTCI is of great importance due to the fact that it is applicable to all seasons and climates together with all spatial and temporal scales.

Human reaction was simulated by the UTCI-Fiala multi-node model of human thermoregulation, which was integrated with an adaptive clothing model. For any combination of meteorological parameters (Eq. 1), based on the conception of an equivalent temperature, the UTCI is the air temperature of the reference environment which, according to the model, produces an equivalent dynamic physiological response under a set of reference conditions (Bröde et al., 2012). In other words, this model simulates the same sweat production or skin wettedness in human body response as the actual environment condition (Błażejczyk et al., 2013; Błażejczyk et al., 2014). This is derived from the multi-dimensional dynamic response of a state-of-the-art multi-node thermo-physiological model of human heat transfer and thermoregulation (Fiala et al., 2012). The UTCI can be represented in general function as bellow:

UTCI = f (t, f, v, $t_{mrt}$)  (Eq.1)

UTCI = f (air temperature, relative humidity, wind speed, mean radiant temperature)

For appropriate use, UTCI can only be approximated using a regression equation abbreviated from sample calculations performed by computing centers (Bröde et al., 2012; Jendritzky et al., 2012). It causes a narrow range of input parameters it

can manage with. For the reference environment, the ISB Commission on the UTCI, was decided to use: (i) a wind speed (v) of =0.5 m·s$^{-1}$ at 10 m height (approximately =0.3 m·s$^{-1}$ at 1.1 m), (ii) a mean radiant temperature ($T_{mrt}$) equal to air temperature, (iii) vapour pressure (vp) that represents a relative humidity (f) of 50%; at high air temperatures (>29°C) the reference humidity was taken to be constant at 20 hPa (Błażejczyk et al., 2013). Physiological parameters, the metabolic rate (M) and the thermal properties of clothing (clothing insulation, permeability) are taken as universal constants in the model due to the evaluation by

means of the regression equation. This implies an outdoor activity of a person walking at the speed of 4 km.h$^{-1}$ (1.1 m.s$^{-1}$), corresponding to heat production of 135 W.m$^{-2}$ (2.3 MET) of metabolic energy (Błażejczyk et al., 2013; Jendritzky et al., 2012) and the clothing insulation which is self-adapting according to the environmental conditions (Havenith et al., 2012). Clothing insulation, vapor resistance and the insulation of surface air layers, are strongly influenced by changes in wind speed and body movement and will therefore also influence physiological responses (Błażejczyk et al., 2012). Particular ranges of UTCI are

categorised according to thermal stress (Table 1).

**Table 1.** UTCI thermal stress classification

*Data and indices considered in the study*

The meteorological data set from the period from 1998 to 2017 was recorded for two urban weather stations (Novi Sad and

Niš) and one rural mountain station (Zlatibor). Mean daily and hourly meteorological parameters (07h, 14h, CET) of air temperature (t), air humidity (f), and wind speed (v) from the above weather stations were considered for the calculation of particular UTCI thermal heat stress in the summer months (July, August and September). The meteorological data set used in the study was retrieved from the Meteorological Yearbook for the period from 1998 to 2017 (Republic Hydrometeorological Service of Serbia) while the UTCI index was calculated by applying the BioKlima 2.6 software package (available at

http://www.igipz.pan.pl/Bioklima-zgik.html).

Although extreme temperatures are one of the most effective climatic parameters in the universal case of hot days and heat waves, the influence of other parameters, especially clothing insulation and metabolic rate, can justify the thermal discomfort and the other way round. In that regard, the summer (July, August and September) daily maximum and minimum temperatures (tmax, tmin), for the same stations, were analysed in terms of particular thresholds to identify the frequency of extremely high

temperatures and heat waves in the period observed to compare them with the discomfort recorded by the UTCI index.

The first stage of the study presented mean daily (UTCI$_{avg}$), morning (UTCI$_{7h}$) and midday (UTCI$_{14h}$) UTCI indices. They are presented by months (July, Aug, Sep) for each station to identify the general differences caused by the geographical origin of the stations under study. The quantification of human bioclimatic conditions in Serbia was designed for the UTCI indices on a daily basis of each defined stress categories. To obtain a better insight into the UTCI index values during the summer months,

averaged monthly UTCI$_{14h}$ index was provided for each year to see how much the UTCI$_{14h}$ index fluctuated in relation to the average monthly value.

For the second stage, midday UTCI$_{14h}$ heat stress indices were identified (strong heat stress SHS, very strong heat stress VSHS, extreme heat stress EHS) and thermal indices based on the maximum and minimum temperatures (hot day HD and hot night HN). During the midday observation (14 CET), particularly prominent are strong heat stress (SHS), which refers to UTCI range of 32°C to 38°C and very strong heat stress (VSHS), which refers to UTCI range of 38°C to 46°C (Table 2).The occurrence of thermal stress days was presented for each month (July, August and September) in the last 20 years. Extreme heat stress (EHS), which refers to UTCI range above 46°C was presented separately when it occurred.

Thermal indices, hot day (HD) and hot night (HN), were identified on the basis of the threshold of the maximum and minimum temperatures. Considering the influence of extreme temperatures, as stated by Collins et al. (2000) the following indices were marked in relation to the thresholds of maximum and minimum temperature: Hot days (HD=$t_{max} \geq 35$°C) and Hot nights (HN=$t_{min} \geq 20$°C) (Table 2). Such a high temperature (35°C or higher), taken as a threshold of what is defined as a heat wave, is assumed to be related to the very strong heat stress category of UTCI index. It should be taken into account that according to the Serbian National Weather Service, a daily maximum temperature of 30°C or higher is taken as the threshold of what is called a tropical day.

For the last stage of the study, with the purpose of identifying thermal discomfort during the period observed, as an illustration of a threshold based on the duration of hot days (HD) and hot nights (HN) a hot day event is determined (HWE =3 HD days of $t_{max} \geq 35$°C). In addition, as an illustration of a threshold based on the duration of very strong heat stress (VSHS), a very strong heat stress event (VSHSE) is determined and it is caused by the occurrence of 5 consecutive VSHS days (Table 2).

As a result of comparing UTCI thermal stress to the selected thresholds of maximum and minimum temperatures, biothermal discomfort was identified during extremely high temperatures for the geographical area of Serbia. The recorded number of days with VSHS stress was compared to the number of days when the maximum air temperature was above 35°C.

**Table 2.** Definition of indices used in the study

### 3. Results

On the basis of the conducted bioclimatic analysis and comparison of biothermal conditions given in the two urban meteorological stations Novi Sad, Niš and Mt. Zlatibor, which represents a rural and lower mountain area, some differences in weather conditions were perceived in the summer period, as was expected. The results are presented in three sections: (i) UTCI indices (UTCI$_{avg}$, UTCI$_{7h}$, UTCI$_{14h}$), (ii) UTCI heat stress indices (strong heat stress SHS, very strong heat stress VSHS, extreme heat stress EHS) and thermal indices (hot day HD, hot night HN), (iii) Heat waves (HWE and VSHSE).

*UTCI indices (UTCI$_{avg}$, UTCI$_{7h}$, UTCI$_{14h}$)*

This section presents the results of the heat budget index UTCI calculated for mean daily data ($UTCI_{avg}$) morning ($UTCI_{7h}$) and midday ($UTCI_{14h}$) for the period of 20 years. In particular, the physiological stress in the category of "strong heat stress" occurred quite frequently in 1998, 2000, 2004, 2007, 2011, 2012, 2015 and 2017 in Niš (July and August) and slightly less frequently in Novi Sad, where "strong heat stress" occurred more frequently in July and August of 2000, 2002, 2006, 2007, 2009, 2013, 2015, 2016 and 2017. A much smaller amount of "strong heat stress" was observed in Mt. Zlatibor. There were periods with the categories of "moderate" and "strong heat stress" evenly distributed in the months of July and August over the period that was investigated. It is important to emphasize that the category of "strong heat stress" most frequently occurred in Mt. Zlatibor in 2000, 2004, 2007, 2012, 2015 and 2017, which generally coincided with the other two stations. These are the years when significant heat waves were recorded. The results depicted in Fig. 2 show the frequency (in percentages) of all stress categories for each index ($UTCI_{avg}$, $UTCI_{7h}$, $UTCI_{14h}$) that occurred in July, August and September for the period that was observed.

**Figure 2**. Monthly frequency of the days with different UTCI stress classes for mean daily ($UTCI_{avg}$), morning ($UTCI_{7h}$) and midday ($UTCI_{14h}$) index during the period from July to September, 1998-2017: a) Niš, b) Novi Sad, c) Zlatibor. X-axis: time (months), y-axis: frequency (number of days)

Bioclimatic conditions in Niš are certainly the most unpleasant ones, given the domination of "strong heat stress" and not so rare "very strong heat stress" categories during the period of 20 years. There are 5 days observed with "extreme heat stress" for midday $UTCI_{14h}$. The most pleasant bioclimatic conditions can be observed in the morning hours ($UTCI_{7h}$), when the dominant categories are "moderate heat stress" (July and August) and "no thermal stress" in September (Fig. 2a). Niš is located in the south of Serbia and belongs to the areas that are most endangered by drought and extreme high temperatures in Serbia. This is confirmed by Tošic and Unkasević (2014) in the study of dry periods in Serbia during the period between 1949 and 2011. It was found that the frequency of droughts in the southern part of Serbia was higher than in the other parts of the country. The most severe drought occurred in Niš and lasted from July 2006 to March 2008 (with a duration of 21 months and severity of 63.0).

Novi Sad is located in the northern part of Serbia, on the slopes of Mt. Fruška Gora and is characterized by a temperate continental and continental climate. Therefore, bioclimatic conditions in Novi Sad are more pleasant than those in Niš. Nevertheless, this does not rule out the occurrence of high temperatures and severe dry periods during the warm part of the year. This idea is based on the findings of Leščešen et al. (2018), when they analyzed drought periods in Vojvodina for over 60 years (1956-2016). All the regions of Vojvodina that were investigated experienced at least one extreme drought event over the reference period, in particular in 2000, 2001-2002, and 2011-2012. Moreover, the same results were obtained in other similar studies (Gocic and Trajković, 2013; 2014; Milanović, et al., 2014). On the mean daily level ($UTCI_{avg}$) the most frequent category is "moderate heat stress" while on the hourly level $UTCI_{14h}$ the most frequent category is "strong heat stress". This is related to the statements made by Di Napoli et al (2018) about two thermal climates in Europe – one of them is associated with

heat stress condition and it is predominant in the southern part of Europe, including the Balkans, when moderate and strong heat stress occurs at central day time hours, which reflects the general relationship between heat load and insolation.

Days with "very strong heat stress" were observed in July and August (Fig. 2b) and there was one day with "extreme heat stress" in July 2007. Mt. Zlatibor has distinctive characteristics of the sub-mountain and mountain climate. Among these three stations, the bioclimatic conditions of Mt. Zlatibor are the most pleasant ones considering the dominant category of "no thermal stress" on the daily level of $UTCI_{avg}$. The thermal conditions with a morning $UTCI_{7h}$ are similar to $UTCI_{avg}$ while the $UTCI_{14h}$ shows the prevailing categories of "moderate heat stress" and slightly less "strong heat stress" and "very strong heat stress". It is important to emphasize the UTCI category of "slight cold stress" for all the indices ($UTCI_{avg}$, $UTCI_{7h}$, $UTCI_{14h}$) was recorded several times during the three months of the period which was observed. The lowest value of the UTCI index was -2.84°C and it occurred on September 5th, 2007 in Mt. Zlatibor as "moderate cold stress" (Fig. 2c).

Of all the calculated UTCI indices, mean daily data ($UTCI_{avg}$), morning data ($UTCI_{7h}$) and midday ($UTCI_{14h}$) data, the midday $UTCI_{14h}$ index shows the most extreme values for identifying heat waves and biothermal heat discomfort. In this regard, for the further data analysis in this study, the $UTCI_{14h}$ is used. To gain better insight into the $UTCI_{14h}$ index values during the summer months, for the investigated period, averaged monthly $UTCI_{14h}$ index for each year was provided for all three stations. The mean monthly values are presented in Fig. 3, together with the trends in $UTCI_{14h}$ and mean absolute deviation of $UTCI_{14h}$ in order to see how much the $UTCI_{14h}$ index fluctuated in relation to the mean monthly value. The results show a significant increase in the extreme values of the $UTCI_{14h}$ index over the last ten years, presented mainly as "very strong heat stress category" (VSHS). A series of peaks can be observed in the years that have been marked as extremely warm. In addition, there is a growing trend of the $UTCI_{14h}$ index in all the summer months and all the weather stations, i.e. a series of peaks with successively higher values during the investigated period (1998-2017). The highest growing trend can be seen in September in Nis and Mt. Zlatibor, while the lowest growing trend can be observed in August on Zlatibor.

As regards the severity of daily UTCI index, there were 5 days recorded in Niš when the $UTCI_{14h}$ exceeded the limit value for "extreme heat stress" (EHS). This happened on July 5th, 2000 ($UTCI_{14h}$=47.08°C); July 24th, 2007 ($UTCI_{14h}$ =48.26°C); August 24th, 2007 ($UTCI_{14h}$ =46.29°C); August 5th and 6th, 2017 ($UTCI_{14h}$ =46.75 and $UTCI_{14h}$ =46.76°C). The maximum value of UTCI in Novi Sad was recorded on July 24th, 2007 (UTCI=48.42°C). In the area of Mt. Zlatibor, the "extreme heat stress" has not been recorded in the last 20 years. The highest values of UTCI in Zlatibor occurred on July 22nd and July 24th, 2007 ($UTCI_{14h}$ =38.62°C and $UTCI_{14h}$ =38.37°C). The year 2007 is rated as the most unfavourable one, particularly the date of July 24th, when the highest temperature ever was recorded in Serbia (Smederevska Palanka, $t_{max}$ = 44,9°C, source: Republic Hydro-meteorological Service of Serbia, RHMSS).

**Figure 3.** The mean monthly $UTCI_{14h}$ index during the period (1998-2017). x-axis: time (years), y-axis: UTCI (°C)

*Thermal indices and UTCI heat stress indices*

This section identifies midday UTCI$_{14h}$ heat stress indices (strong heat stress SHS, very strong heat stress VSHS) and thermal indices marked in relation to the thresholds of the maximum and minimum temperatures in particular (hot day HD and hot night HN). Fig. 4 presents frequency (number of days) of hot days (HD), hot nights (HN), strong heat stress (SHS) and very strong heat stress (VSHS) for each month (July, August and September) in the last 20 years. The results of UTCI that were obtained show compatibility with previous analyses in relation to the occurrence of heat waves, especially since 2000 (Basarin et al., 2018; Unkašević and Tošić, 2009a; Unkašević and Tošić, 2009b; Unkašević and Tošić, 2013).

In accordance with the thresholds of maximum temperatures in July, there were several years in Niš with 10 or more HDs in July, when the temperature was above 35ºC, as shown in Fig. 4. This can be observed in July of 2000, 2007, 2012, 2015, and 2017, and in total, for each year it amounts to 10, 18, 16, 11 and 11 HD days, respectively. As regards the thresholds of maximum temperature in Novi Sad, there were 10 or more HD days with temperatures above 35ºC, more specifically 10 and 11 HD days during July of 2012 and 2015, respectively. In Mt. Zlatibor only two HD days were recorded in July 2007. In August, this can be observed in 2000, 2007, 2012, 2013, 2015, and 2017 and in total it amounts to 12, 12, 10, 10, 13 and 13 HDs days respectively for each year in Niš. The highest number of days with a temperature over 35ºC degrees was recorded in 2000, 2012, 2015 and 2017 and in total for each year it amounts to 9, 8, 11 and 13, respectively. In Mt. Zlatibor there are no HD days recorded in August (Figure 4).

**Figure 4.** Number of hot days (HD), hot nights (HN), strong heat stress (SHS) and very strong heat stress (VSHS) in Niš, Novi Sad and Zlatibor (July, Aug, Sep) 1998-2017. x-axis: time (years), y-axis: number of days

For the period investigated (1998-2017), the total number of hot days recorded in July, August and September in Niš was 126, 133 and 21 HDs, respectively, while Novi Sad saw about a half of the said number of hot days, namely 52, 63 and 3 HDs, respectively. In Mt. Zlatibor 2 HD days were recorded in July.

As regards hot nights (HN), there is generally a lower intensity of HNs compared to HDs. Generally, during the observed period up to 4 HNs were recorded, although an increase in the HNs has been observed in the last ten years. For example, in Niš during 2012, 2015 and 2017 the number of HNs recorded in July was 12, 7 and 7 respectively and during 2000, 2013, 2015, and 2017 the number of HNs recorded in August was 8, 6, 5 and 5 respectively. Since 2010 up to 2 HNs have been recorded in September. In Novi Sad, during 2010, 2012, 2015 and 2017 there were 7, 5, 5 and 6 HNs respectively recorded in July, while in August during 2012 and 2017 there were 5 and 7 HNs respectively. In September, one HN was recorded during 2009 and 2011 (Figure 4). The total number of hot nights recorded in July, August and September for the investigated period in Niš was 60, 56 and 6 HNs, respectively, while in Novi Sad there was a significantly lower number of hot nights – 41, 38 and 2 HNs, respectively. In Mt. Zlatibor 9 HN days were recorded in July and 11 HN days were recorded in August for the period which was investigated.

Midday UTCI$_{14h}$ "strong heat stress" (SHS) occurs most commonly in all three stations. For example, SHS is particularly important to Mt. Zlatibor, located at an altitude above 1000 m since there has been an increase in the number of days with SHS in Zlatibor over the last decade. The best indicator is the increase in SHS in September. Similarly, the indicator of biothermal discomfort, "very strong heat stress" (VSHS) has been occurring more frequently in the last decade, reaching a maximum of

17 days in July 2007 and 2012 and in August 2015 in Niš. The number of VSHS days in Novi Sad reached a maximum of 12 days in July 2015 and 14 days in August 2015 and 2017. The total number of VSHS days recorded in July, August and September for the investigated period in Nis was 167, 174 and 25 VSHS days, respectively, while in Novi Sad there was a significantly lower number of VSHS days – 83, 97 and 7 VSHS days, respectively. In Mt. Zlatibor 2 VSHS days were recorded in July and 3 VSHS days were recorded in August. The recorded numbers of VSHS days indicate slightly greater biothermal

discomfort duration in relation to the number of HD days. The HD and VSHS indices are significantly correlated during the summer months as they are directly derived from the maximum air temperatures and UTCI$_{14h}$ index (Table 3). UTCI$_{14h}$ and $t_{max}$ for the Niš and Novi Sad urban stations have a significantly high linear correlation.

**Table 3.** Linear correlation of UTCI$_{14h}$ and $t_{max}$

For Zlatibor the correlation is slightly lower in July and August, while in September is weak compared to the urban stations, but neither HD nor VSHS indices occurred on Zlatibor. Similar results of the linear relationship between the air temperature and the UTCI index (r = 83; r = 0.98) were shown in the studies of Urban and Kysely (2014) for summer in the urban areas of South Bohemia in Czech Republic and in the studies of Nassiri et al. (2017) for Iran.

*Heat waves (HWE and VSHSE)*

After UTCI thermal stress was compared to the selected thresholds of maximum temperature, biothermal discomfort was identified during extremely high temperatures for the geographical area of Serbia. During the observed period, a heat wave event (HWE) was determined, caused by 3 consecutive HD days and very strong heat stress event (VSHSE), caused by the

occurrence of 5 consecutive VSHS days.

Table 4 lists the identified heat waves that refer to UTCI$_{14h\ VSHS}$ index (VSHSE) and HD index (HWE). It has a very similar layout. However, shadowed differences can be observed, especially in the duration of the heat wave event. The results are presented for Niš and Novi Sad since there were no recorded heat waves in Mt. Zlatibor. More heat waves, both HWEs and VSHSEs, have been identified in the last ten years.

A heat wave event (HWE) of 10 days in a row (Niš, July from 15[th] to 24[th], 2007) is the maximum number of consecutive days with such high temperatures when all the three stations are compared. In the same month, HWE of 3 days in a row (July 8[th] to 10[th]) and 4 days in a row (July 27[th] to 30[th]) were observed (Table 4, HWE). Altogether, the heat wave events (HWEs) occurred

3 times and they lasted 3, 10 and 4 days in July 2007, which is certainly an extreme for the observed period. The heat wave events (HWE) which lasted 3 or 4 days in a row occurred during 9 years in the observed period of 20 years. From that source, heat wave events (HWE) occurred twice and lasted 3 to 4 days during July in 2004 and 2017 (Table 4). It is necessary to point out that heat waves in July 2002, 2005, 2007, 2012 and 2015 lasted more than five HD days in a row with $t_{max}$ above 35°C, amounting to 6, 6, 10, 9 and 7 HD days respectively. During August heat wave events (HWE) with 3 to 6 hot days in a row occurred twice in 1999, 2000, 2001, 2012, 2015 and 2017. Furthermore, in August, series of heat wave events with 5 hot days in a row or more were observed in 1998, 2000, 2007, 2012, 2013, 2015 and 2017 and it amounts to 5, 6, 6, 6, 7 and 6 hot days in a row, respectively. As regards the thresholds in Novi Sad, there were two heat waves events (HWE) with 3 and 4 hot days in a row which occurred in August of 2012 and 2015. In 2017 3 heat wave events (HWE) occurred with 6, 3, and 3 hot days in a row. It is certainly important that two HWEs occurred during September 2015 and lasted 3 and 5 HD days in a row with $t_{max}$ above 35°C (Table 4). This justifies the finding of Tomczyk (2016) that the heat waves began to occur in September in the last decade. As shown in Fig. 4, the number of HDs in September, during the period of 20 years increased significantly after 2007, especially in Niš.

**Table 4.** Number of Heat waves events (HWE) and Very strong heat stress event (VSHSE) in Niš, Novi Sad - July, Aug and Sep, 1998-2017

Of the total number of hot days, it amounts to 21 HDs in September in Niš, only one HD was recorded in the first decade (1998-2007) of the investigated period, while the remaining HDs were recorded over the last decade (2008-2017), especially in 2011 and 2015. In the second decade of the investigated period the year 2015 stands out with two HWEs (September 1st to 5th, and 17th to 19th). In the same year the highest daily temperature was recorded in September (on September 18th, $t_{max}$=37.5°C). A similar situation occurred in the area of Novi Sad. During the first 10 years, not a single HD was recorded in September while after 2007, 5 such days were recorded. For Niš and Novi Sad year 2015 stands out and the hottest September days were September 17th and 18th with $t_{max}$=36.7°C and 36.4°C.

For both weather stations in the second decade, together with the increase in the number of HDs, there was also an increase in the number of HNs. In September, the daily air temperature increases, so days with temperatures around 30°C and higher are more frequent. In this weather station year 2015 stands out with the same date of the hottest day in September (September 18th, tmax=33.2°C). This correlates with the heat waves analysis in Athens, where period of heat wave lasted from mid-June to the beginning of September (Papanastasiou et al., 2014). In Mt. Zlatibor area there was no significant phenomenon of extreme temperatures and therefore no HWEs were recorded.

Bioclimatic conditions analyzed by means of the UTCI index show that the calculated data observed at 14 CET are related to the marked heat wave events (HWE). When $UTCI_{14h}$ thermal stress category of "very strong heat stress" (VSHS) is compared to the selected heat wave events (HWE,) a sub-index is defined. It is called very strong heat stress event (VSHSE) and it is

caused by the occurrence of 5 consecutive VSHS days. The VSHSE corresponds to heat wave event and provokes a severe biothermal discomfort, so it was used as an indicator of extremely unfavorable bioclimatic conditions (biothermal heat hazard). Along these lines, the VSHSE in Niš occurred during July in 1998, 2004, 2005, 2007, 2012 and 2015 where 7, 5 (on two occasions), 6, 10, 12 and 10 VSHS days in a row were recorded respectively (Table 4, VSHSE). In August, VSHSE was recorded on 5 VSHS days in a row in 1998, 1999, 2008, 2010, 2012 and 2017, while in 2000, 2012, 2013, 2015 and 2017 6, 6, 8, 7 and 6 VSHS days in a row were recorded respectively. It should be emphasized that there were two VSHSE events in 2012 and 2017, for each year 5 and 6 VSHS days in a row. September 2015 saw a VSHSE with 5 VSHS days in a row. According to Unkašević and Tošić (2009b) the highest temperatures in Niš ever were recorded during the summer of 2007 (44.2°C) and the summer of 2000 (42.5°C), covering the data period from 1948 to 2007. The VSHSE occurred in Novi Sad in July 2007 (7 VSHS days in a row), 2012 (6 VSHS days in a row) and 2015 (8 VSHS days in a row) and in August 2000 (7 VSHS days in a row), 2013 (8 VSHS days in a row), 2015 (6 VSHS days in a row) and 2017 (6 VSHS days in a row). In August 2012 VSHSE occurred twice and each time 5 VSHS days in a row were recorded. There was no VSHSE recorded in Zlatibor.

## 4. Discussion

The purpose of this study was to investigate biothermal condition in Serbia during summer and provide a comprehensive assessment of human heat stress connected with UTCI index during heat waves. The results obtained in the study indicates the general increase in biothermal discomfort associated with heat waves defined by maximum air temperature above 35°C and UTCI "very strong heat stress" above 38°C. Morning, midday and average UTCI show significant occurrence of thermal heat stress (moderate, strong and very strong heat stress) during the summer months and increase in biothermal discomfort especially for $UTCI_{14h}$. This is confirmed by obtained distribution of the average monthly $UTCI_{14h}$ showing a significant increase in the extreme values of the $UTCI_{14h}$ index in the last ten years, presented mainly as "very strong heat stress category" (VSHS) together with a growing trend of the $UTCI_{14h}$ index in all the summer months for all three investigated weather stations, primarily in Niš. These findings show that heat waves and biothermal heat discomfort may be more frequent and longer in the future and this indicates the need for biothermal heat warnings. The increase in extreme biothermal heat conditions is the most evident in the number of days with UTCI index thresholds between 38°C and 46°C, defined in the study as "very strong heat stress" (VSHS) and maximum air temperature thresholds above 35°C, defined in the study as hot days (HD). Thus, most HDs (18 days) occurred in 2007 and then in 2012 (16 days) means that more than half of July in these years was characterised by extremely high temperatures. Of particular importance is the increasing number of HD, HN, SHS and VSHS days in September since 2008 (Fig 4). Similar results about increase in daily maximum air temperature were obtained by Unkašević and Tošić (2011), when the record values of the maximum temperatures were observed for almost the whole territory of Serbia in 2007. As reported by Papanastasiou et al. (2014), the summer of 2007 was the warmest summer in Athens in the last hundred years. This follows the statement made by Di Napoli et al. (2018) about UTCI reference value from the period between 1979 and

2016, when the UTCI at 12:00 (UTC) was about 0.5°C colder in the period between 1980 and 1999, while it was 0.5°C warmer in the period between 2000 and 2009 and 1°C warmer in the period between 2010 and 2016.

The significant findings from this study imply that UTCI category of "very strong heat stress" (VSHS) defines extreme biothermal heat discomfort that can be considered biothermal heat hazard. The $UTCI_{14h}$ category of "very strong heat stress" (VSHS) correlates with the "hot day" (HD) and there has been an increase in the number of such days over the last ten years. Particularly severe biothermal heat discomfort occurred in 2007 (18 VSHS days) and 2012 (17 VSHS days). Nevertheless, the total number of VSHS days (558 days) in all three stations for investigated period is greater than the total number of HD days (400 days) which indicates a slightly longer duration of biothermal heat discomfort defined by the UTCI index compared to number of HD days. This is important because there is a difference between the biothermal heat stress defined by VSHS and potential heat stress defined by maximum temperatures where the calculation of UTCI, apart from temperature, includes some other parameters such as humidity, wind speed and direct solar radiation, together with metabolic rate and clothing insulation. Extremely high temperature does not necessarily cause heat stress if the wind is strong, and the relative humidity is low or if a person is not being physically active. For instance, the situation in Novi Sad on July 30th 2007, when the maximum temperature was higher than the calculated UTIC index ($T_{max}$=26.9°C; $UTCI_{14h}$=15.7°C, no thermal stress) does not necessarily indicate biothermal heat stress if the wind speed is high (6.7 m.s$^{-1}$) or the relative humidity is lower (52%). On the other hand, the situation on August 26th 2002, when the maximum air temperature was lower than the calculated UTIC index ($T_{max}$ = 24.5°C; $UTCI_{14h}$ = 30.6°C) indicates that heat stress may still occur if the wind speed is low (0.8 m.s$^{-1}$) or the relative humidity is higher (57%). This indicates the sensitivity of the index to environmental parameters, especially wind speed, humidity, air temperature and solar radiation. In this sense, determining extreme weather event from the aspect of human heat budget indices allows other meteorological parameters to be taken into account. Apart from temperature and humidity, wind and direct solar radiation, have been considered together with metabolic rate and clothing insulation.

The identified heat waves based on the threshold of the UTCI index correspond to the identified heat waves based on the threshold of maximum air temperatures. Furthermore, there is a significant increase in heat waves (HWE and VSHSE) plus duration of such events over the last ten years in Niš and Novi Sad, especially in Niš. Similar results about increase of heat waves in Novi Sad defined by other heat budget index Physiological Equivalent Temperature (PET) for the period from 1949 to 2012 reported Basarin et al. (2016) where the highest number of heat waves was observed in the last two decades and in the first decade of the investigated period, while the number of the days above particular thresholds for the period 1961-2014 shows an increase along with the number of heat waves per year since 1981. The urban areas have a high risk of heat absorption of buildings and asphalt, which results in the formation of high temperatures during the night (Giannopoulou et al., 2014). Rural and mountainous areas rich in forests and greenery are less exposed to these phenomena, like Mt. Zlatibor, but during the hottest summer months, days and nights with high temperatures can occur (Fig. 2c).

It is important to point out that several heat waves occurred in summer. The analysis highlights 2007, 2012, 2015 and 2017 as the years with the most heat waves registered as heat wave events (HWE), with special emphasis on Niš, where the longest HWE lasted for 10 days in July, 2007 with UTCI index over 38°C and maximum temperature over 35°C (Table 4). This agrees

with the findings of Unkašević and Tošić (2011, 2015) that there has been a growing trend of heat waves in Serbia, especially since 2000. In order to justify the present research for the last 20 years, the earlier results related to the increase in heat waves will be discussed in more detail. In that regard, previous research into heat waves in Niš for the period from 1949 and 2007, based on the autoregressive-moving-average model, observed the warmest years during three periods, 1951-1952, 1987-1998 and 2000-2007 (Unkašević and Tošić, 2009b). According to this research, the longest heat wave was recorded in Niš in 1952, with 21 days, while in 2003, 29 consecutive tropical days were observed. It should be taken into account that according to the Serbian National Weather Service, tropical days are days with a maximum temperature over 30ºC. Furthermore, based on Heat Wave Duration Index (HWDI) i.e. daily maximum values of air temperature, Drljača et al. (2009) determined the duration and strength of heat waves in Niš during the summer season. The analysis showed Niš with a greater number of heat waves compared to the larger urban area of Belgrade. As stated in the research, since the mid-1980s, heat waves have had a higher frequency and on average they have occurred every year. Prior to that period heat wave fluctuations were generally reported in one of two years (Niš during the summer) (Drljača et al., 2009). While studying the characteristics of the heat waves in central Serbia (1949-2007), Unkašević and Tošić (2011) detected an increase in heat wave duration, in addition to an increase in heat wave frequency of occurrence during the period from 1999 to 2007. Besides, based on the HWDI index, heat waves in Novi Sad show a trend of increasing tropical days ($T_{max} \geq 30.0°C$) from 1960 to 2010, indicating years 1994, 1998, 2000, 2003, 2007, 2009 (Malinović-Milićević et al., 2013).

Further, 2 heat wave events (HWE) were highlighted in September 2015 and they lasted 5 and 3 days, as well as a very strong heat wave event (VSHSE) which lasted 5 days. This agrees with Tomczyk (2016) statement that in the last decades, heat waves have begun to occur in September in south-eastern Europe, where usually the highest number of HWs recorded in July and August.

The biothermal discomfort identified in July and August as the hottest summer months in Serbia, could provoke health disorders more frequently. Human sensitivity to extreme heat weather can also be seen in the impact of heat waves on daily mortality in Belgrade during the summer, when the strong correlation between heat waves and daily mortality can be observed in July 2007 (Stanojević et al., 2014). Similarly, Bogdanović et al. (2013) reported a significant short-term excess mortality on 16[th] July in Belgrade, when the maximum daily temperature exceeded 35°C, leading to 167 excess deaths (38% increase compared to the number of expected deaths) for nine consecutive days of heat, with a progressive return to almost normal mortality as the maximum temperature dropped below 35°C on 25[th] July. Specific HWs characteristics such as intensity and duration may have devastating effects on human health and wellbeing. In certain cases, heat waves cause problems to children, elderly people, chronic patients and workers are particularly susceptible to them.

However, this study probably has potential limitations. The period under observation could be longer for the purposes of keeping track of the extremes trend. This shortcoming was attenuated by reference to a more detailed review of previous studies on the trend of maximum temperatures in Serbia reported by Unkašević and Tošić (2009a, 2011, 2015), Drljača et al., 2009, Malinović-Milićević et al. (2013, 2016). Furthermore, complete hourly weather data are missing and therefore, they are not sufficient for a more detailed analysis of UTCI values, which might be used for healthcare purposes. As for the UTCI

index, it should be emphasized that the model is limited by a fixed metabolic rate that approximates light physical activity (1.1ms$^{-1}$). In other words, biothermal heat discomfort can cause a stronger heat load in humans if the physical activity is more intense. Considering wide spatial dimension of diverse geographic and climate regions where UTCI has been applied, however, adaptation to existing biothermal conditions, hot or cold, should be excluded.

Nevertheless, the research results of this study highlight the importance that UTCI as a bioclimatic indicator has for biothermal heat discomfort, particularly if it is connected with the effects on human health. This study features a human bioclimatic method in analysing a biothermal condition with special emphasis on heat stress, so that the impact on human health and well-being can be understood better.

The application of the standardized bioclimatic heat budget index UTCI help improve the understanding of biothermal conditions related to heat wave. Obtained results from this study show that UTCI$_{14h}$ heat stress becomes more extreme in terms of severity and heat wave duration up to very strong heat stress (VSHS). The biothermal indices investigated for the three weather stations follow the trend of general warming. As reported by Vuković et al. (2018), future change analysis in Serbia, concerning the base period 1986-2005, in compliance with Intergovernmental Panel on Climate Change (IPCC) fifth assessment report (AR5), shows the increase in temperature by the end of the 21st century, which proves that it is necessary to take immediate measures to alleviate negative impacts. In the light of climatic changes and other negative factors resulting from this global phenomenon, it is becoming a true challenge to minimize their effects and improve living conditions in urban and rural areas (Stevović et al., 2017). Bearing in mind the above, the findings of this study indicate the need to make plans regarding adaptation and mitigation measures in public weather services, public health systems, urban planning, tourism and recreation and climate impact research. This implies creating high density network of measuring urban stations for potential monitoring of biothermal heat discomfort as long as UTCI has application in weather forecast regarding outdoor activities, appropriate behaviour and climate therapies, and extreme thermal stress alerts. For future research, it is necessary to expand spatial analyses in the region of Balkans. However, the human heat budget indices are increasingly serving as a determinant of extreme weather condition (Di Napoli et al., 2018; Matzarakis et al., 2018; Theoharatos et al., 2010; Urban and Kyselý, 2014). UTCI index improves the understanding of extreme thermal stress, serving as one of the criteria for initiating heat alerts about extreme thermal stress in Serbia.

### 5. Conclusion

The present research of the human bioclimatic evaluation of UTCI thermal stress in Serbia is considered important due to the identification of biothermal heat hazard and the study of the evolution of such indices regarding climate change. This evaluation aims at providing a comprehensive assessment of the human heat stress associated with UTCI index and heat waves defined by UTCI very strong heat stress and maximum air temperature.

The assessment of human biothermal conditions for the investigated period from 1998 to 2017 was provided for three synoptic stations – two urban (Niš and Novi Sad) and one station representing mountain areas of an altitude up to 1500 m (Zlatibor).

All of them are located in different geographical areas in Serbia. It was found that morning, midday and average UTCI show significant occurrence of all heat stress categories during the summer months and increase in biothermal discomfort especially for $UTCI_{14h}$. The findings in the study show an increase in biothermal discomfort associated with heat waves defined by $UTCI_{14h}$ "very strong heat stress" (VSHS) above 38°C and heat waves defined by maximum air temperature above 35°C. The VSHS describes an alarming biothermal state and has occurred frequently in the last ten years, particularly in Niš and Novi

Sad. The findings indicate $UTCI_{14h}$ index "very strong heat stress event" (VSHSE) as an indicator of biothermal heat hazard corresponding to heat waves defined by maximum air temperature. The most extreme heat waves (HWE and VSHSE) occurred in 2007, 2012, 2015 and 2017. Heat wave events (HWE) lasting 10 days in July 2007, 9 days in July 2012 and heat wave events (VSHSE) lasting 10 days in July 2007 and July 2015 and those lasting 12 days in July 2012, which occurred in Nis, are the events with maximum duration. The fact that heat waves occurred twice in September in 2015 is certainly an important

finding. Undoubtedly, heat wave events are one of the natural hazards with increasing impact in urban areas related to higher population density. Considering these facts, it can be deduced that heat waves are becoming more frequent, stronger and longer. Thus, frequent heat waves since 2007 with UTCI above 38°C certainly indicate biothermal heat hazard. The longer events of bioclimatic discomfort could indicate more stressful effect on bioclimatic conditions for human health and wellbeing. However, the results of the study highlight the importance of UTCI index as a bioclimatic indicator of biothermal hazard in

Serbia particularly if it can serve as one of the criteria for initiating heat warnings in public weather services, public health systems, urban planning, tourism and recreation and research into climate impact in Serbia.

**Acknowledgements:** The paper represents the results of research on the National projects supported by Ministry of Education, Science and Technological Development, Republic of Serbia (No. III 47007; 176017 and 176008). The authors highly

appreciate comments and suggestions of reviewers.

**Author's contribution:** MP designed the study, collected, analysed and interpreted the data, wrote the manuscript and processed the figures. ML collected the data, supported the writing and processed the figures. DF carried out the management activities. BP participated in the technical support. UB contributed to the interpretation of the results and the improvement of the figures All authors contributed to

the discussion and interpretation of the results.

**Conflict of interest:** We claim that all authors agree to the submission of the manuscript. This manuscript has not been published before and is not concurrently being considered for publication elsewhere. This manuscript does not violate any copyright or other personal proprietary right of any person or entity and it contains no statements that are unlawful in any way.

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

**Table 1.** UTCI thermal stress classification

| UTCI (°C) | > 46 | 38 to 46 | 32 to 38 | 26 to 32 | 9 to 26 | 0 to 9 | 0 to -13 | -13 to -27 | -27 to -40 | < -40 |
|---|---|---|---|---|---|---|---|---|---|---|
| Stress classes | extreme heat stress | very strong heat stress | strong heat stress | moderate heat stress | no thermal stress | slight cold stress | moderate cold stress | strong cold stress | very strong cold stress | extreme cold stress |
| Abbr. | EHS | VSHS | SHS | MHS | NTS | SLCS | MCS | SCS | VSCS | ECS |

Source: Błażejczyk et al. 2013






**Table 2.** Definition of indices used in the study

| Abbrevations | UTCI Indices | Definition |
|---|---|---|
| $UTCI_{avg}$ | Universal Thermal Climate Index | $UTCI_{avg} = f\,(t,\,f,\,v,\,t_{mrt})$ |
| $UTCI_{07h}$ | Universal Thermal Climate Index$_{07h}$ | $UTCI_{07h} = f\,(t_{7h},\,f_{7h},\,v_{7h},\,t_{mrt})$ |
| $UTCI_{14h}$ | Universal Thermal Climate Index$_{14h}$ | $UTCI_{14h} = f\,(t_{14h},\,f_{14h},\,v_{14h},\,t_{mrt})$ |
| Abbrevations | UTCI heat stress indices | Definition |
| SHS | Strong Heat Stress | SHS= UTCI (32ºC to 38ºC) |
| VSHS | Very Strong Heat Stress | VSHS = UTCI (38ºC to 46ºC) |
| VSHSE | Very Strong Heat Stress Event | minimum 5 VSHS days in a row |
| EHS | Extreme Strong Stress | EHS= UTCI >46ºC |
| Abbrevations | Thermal indices | Definition |
| HD | Hot days | $t_{max} \geq 35°C$ |
| HN | Hot nights | $t_{min} \geq 20°C$ |
| HWE | Heat waves event | minimum 3 HD days in a row |

t-air temperature; f-relative humidity; v-wind speed; tmrt-mean radiant temperature; tmax-maximum air temperature, tmin-minimum air temperature



Table 3. Linear correlation of $UTCI_{14h}$ and $t_{max}$

| $UTCI_{14h} \sim t_{max}$ | Niš | Novi Sad | Zlatibor |
|---|---|---|---|
| July | 0.88 | 0.97 | 0.85 |
| August | 0.92 | 0.95 | 0.77 |
| September | 0.89 | 0.92 | 0.41 |







**Table 4.** Number of Heat waves events (HWE) and Very strong heat stress event (VSHSE) in Niš, Novi Sad -  July, Aug and Sep, 1998-2017

| HWE | July | | Aug | | Sep | VSHSE | July | | Aug | | Sep |
|---|---|---|---|---|---|---|---|---|---|---|---|
| | Niš | Novi Sad | Niš | Novi Sad | Niš | | Niš | Novi Sad | Niš | Novi Sad | Niš |
| 1998 | 4 | / | 5 | / | / | 1998 | 7 | / | 5 | / | / |
| 1999 | / | / | 3, 4 | / | / | 1999 | / | / | 5 | / | / |
| 2000 | 3 | / | 4, 6 | 6 | / | 2000 | / | / | 6 | 7 | / |
| 2001 | 3 | / | 3, 3 | / | / | 2001 | / | / | / | / | / |
| 2002 | 6 | / | / | / | / | 2002 | / | / | / | / | / |
| 2003 | 4 | / | / | / | / | 2003 | / | / | / | / | / |
| 2004 | 3, 4 | 3 | / | / | / | 2004 | 5,5 | / | / | / | / |
| 2005 | 6 | / | / | / | / | 2005 | 6 | / | / | / | / |
| 2006 | / | / | 3 | / | / | 2006 | / | / | / | / | / |
| 2007 | 3, 10,4 | 6 | 6 | / | / | 2007 | 10 | 7 | / | / | / |
| 2008 | / | / | 4 | 3 | 3 | 2008 | / | / | 5 | / | / |
| 2009 | / | / | 3 | / | / | 2009 | / | / | / | / | / |
| 2010 | / | / | 4 | / | / | 2010 | / | / | 5 | / | / |
| 2011 | 3 | 4 | 3 | / | / | 2011 | / | / | / | / | / |
| 2012 | 9 | 6 | 4, 6 | 3, 5 | / | 2012 | 12 | 6 | 5,6 | 5,5 | / |
| 2013 | / | / | 7 | 4 | / | 2013 | / | / | 8 | 8 | / |
| 2014 | / | / | / | / | / | 2014 | / | / | / | / | / |
| 2015 | 7 | 9 | 6, 4 | 4, 5 | 5, 3 | 2015 | 10 | 8 | 7 | 6 | 5 |
| 2016 | 3 | / | / | / | / | 2016 | / | / | / | / | / |
| 2017 | 4, 4 | / | 6, 4 | 6, 3, 3 | / | 2017 | / | / | 6,5 | 6 | / |

Note: pink cells have more than one HWE and VSHSE duration defined by numbers of the consecutive days






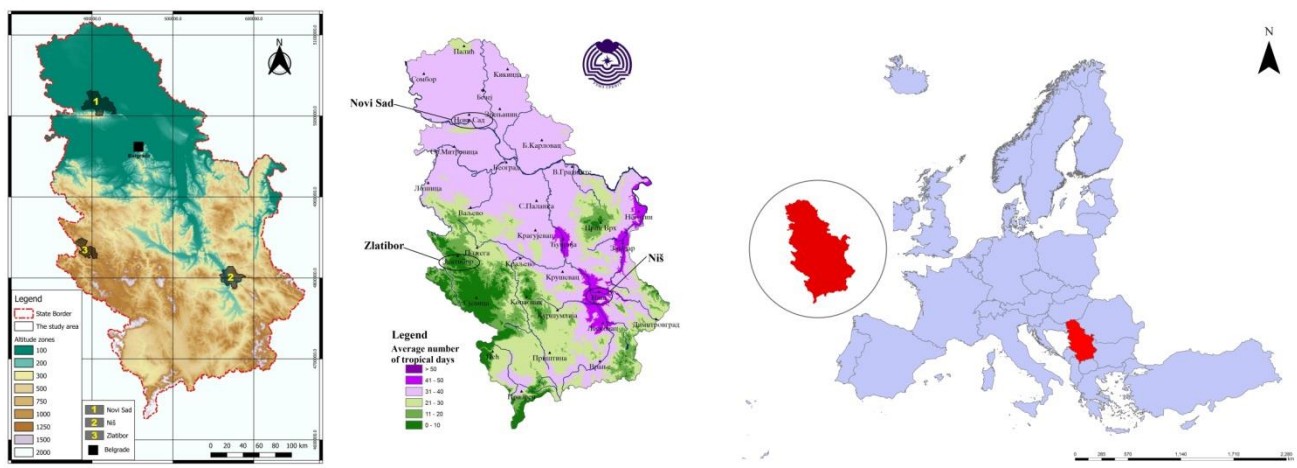


**Figure 1**. Relief map of Serbia with the studied areas: 1-Novi Sad, 2-Niš, 3-Zlatibor (left). Map of the average number of tropical days in Serbia per year for the period 1981-2010 (middle) (source: Republic Hydro-meteorological Service of Serbia RHMSS) and a map of geographical location of Serbia in Europe (right) [Maps were created using QGIS 3.8 software on the basis of the official European Commission's (Eurostat) datasets, available at 860 https://ec.europa.eu/eurostat/web/gisco/geodata/reference-data/, (Map ratio: 1:1900000; Map projection: WGS 84/UTM, Zone 34N, the official national coordinate system)].



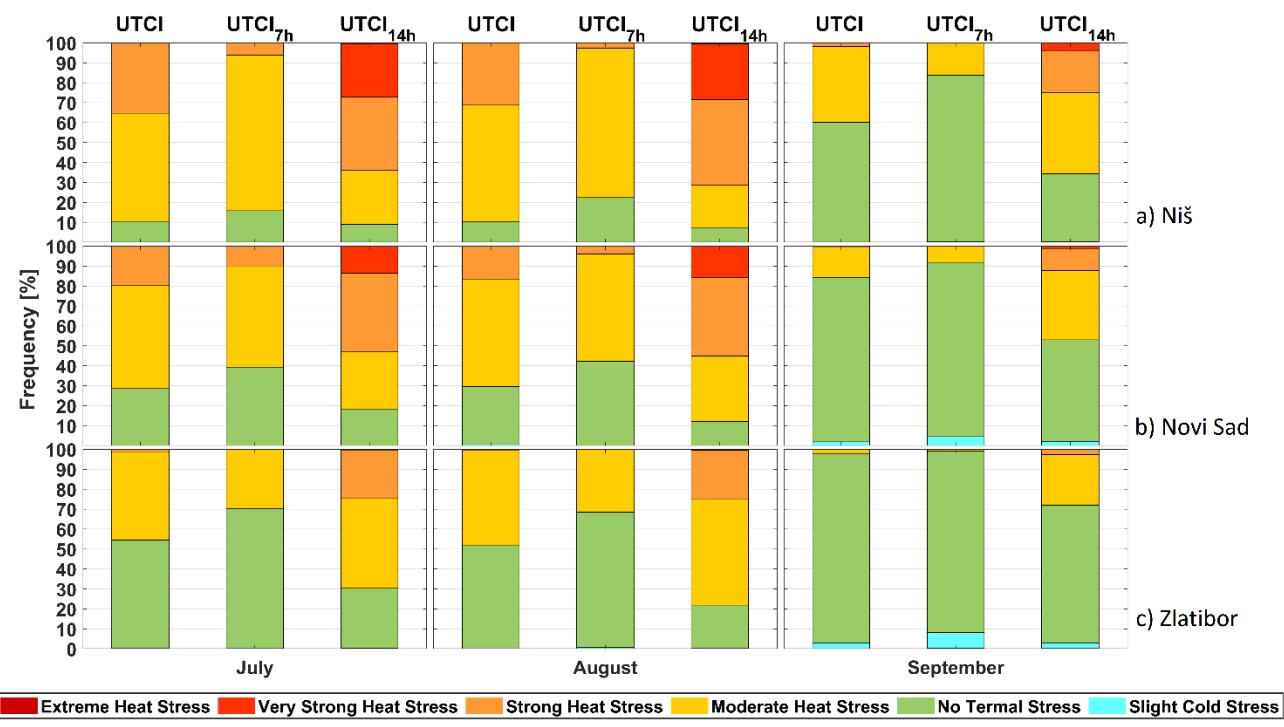

**Figure 2**. Monthly frequency of the days with different UTCI stress classes for mean daily (UTCI$_{avg}$), morning (UTCI$_{7h}$) and midday (UTCI$_{14h}$) index during the period from July to September, 1998-2017: a) Niš, b) Novi Sad, c) Zlatibor. X-axis: time (months), y-axis: frequency (number of days)



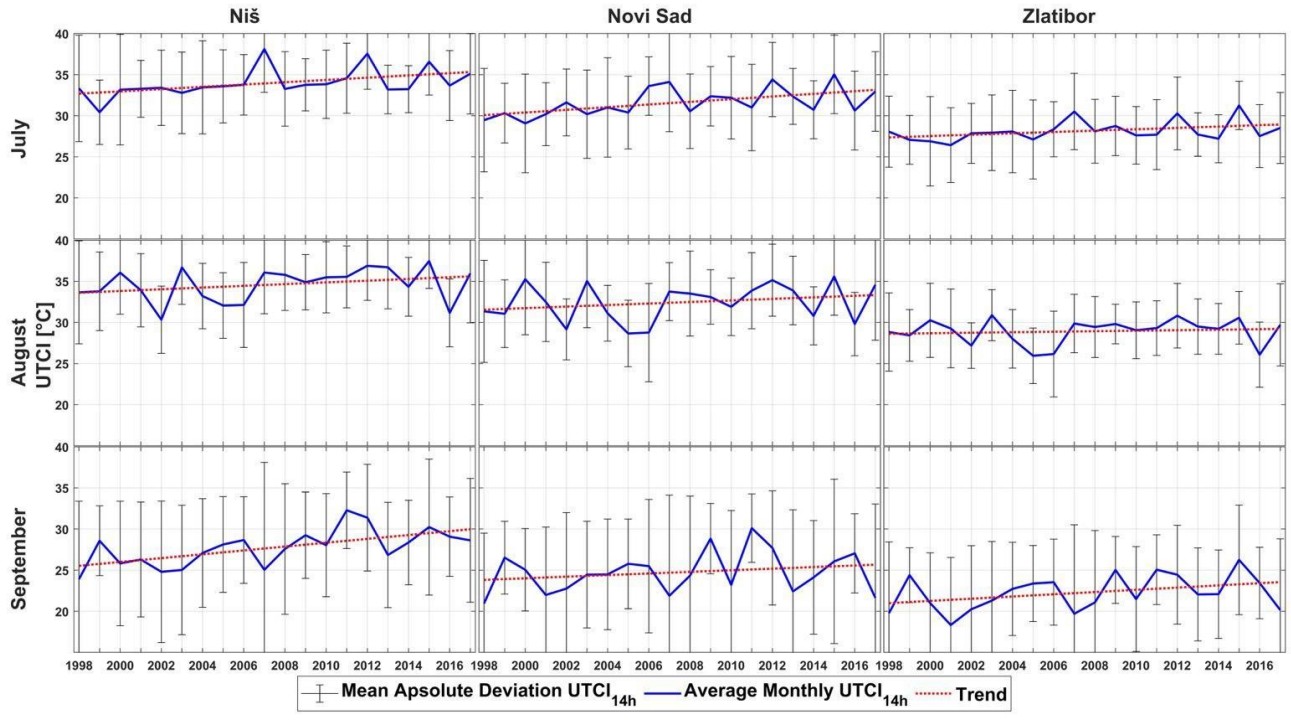

**Figure 3.** The mean monthly UTCI$_{14h}$ index during the period (1998-2017). x-axis: time (years), y-axis: UTCI (°C)



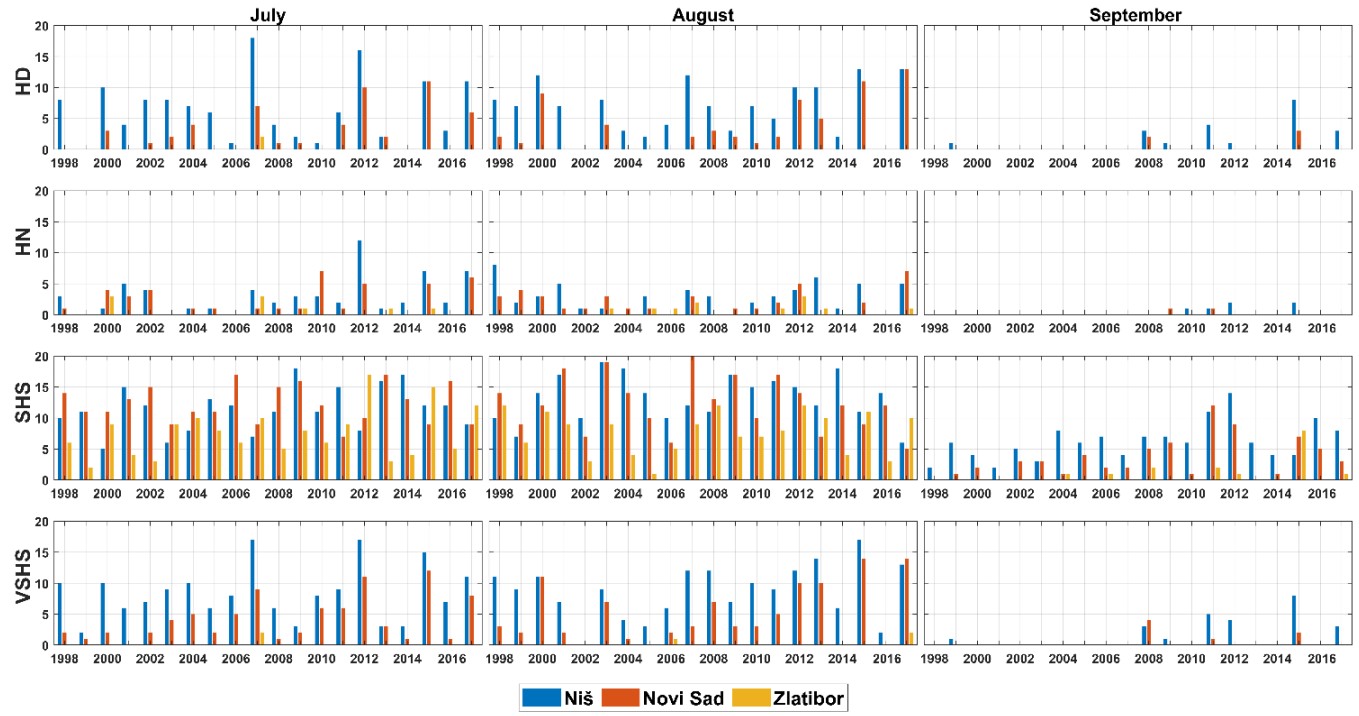

**Figure 4.** Number of hot days (HD), hot nights (HN), strong heat stress (SHS) and very strong heat stress (VSHS) in Niš, Novi Sad and Zlatibor (July, Aug, Sep) 1998-2017. x-axis: time (years), y-axis: number of days
