# Peer review of "Analysis of UTCI index during heat waves in Serbia"

_Natural Hazards and Earth System Sciences, 2019_

## Referee Comment (RC1) · Anonymous Referee #1 · 9 Dec 2019

The paper presents an assessment of bioclimatic conditions in Serbia. A bioclimatic Universal Thermal Climate Index (UTCI) is computed for three weather stations over the last twenty years. Daily maximum temperatures are also analysed in order to identify increase and frequency of heat waves in Serbia. This work could potential be interesting however improvement of the paper is need previous to publication in order to reach the expected international standards requested by the journal. First of all, according to the elements presented in the introduction, many similar studies have already been conducted in Serbia. Within the current study, a more recent meteorological data sets are used to compute the UTCI index values for the selected stations. Starting before 1998 would have been even more interesting to identify the trend of such extreme heat waves.

[Figure]

Title To me the title could be:" Analysis of UTCI index during heat waves in Serbia"

Abstract/introduction The abstract is concise and seems complete. However, the introduction needs to be improved. The first part introduces the UTCI index, then heat waves in Serbia are presented, followed by recent studies (line 75 to 94). This final part of the introduction is difficult to understand. It is better to rewrite it and introduce that there is a need to study the evolution of such indices regarding climate change, as trend has been identified during the period 1999-2007.

Materials and methods The model used to compute the index is not explained and presented. The reader does not know how the index are computed and therefore cannot judge the methodology and evaluate the results. Therefore, within the material and method section, it is needed to add the equations of the model. Moreover, to me a validation of the model should be introduced in the section, this will give more weight to the results, moreover this will help the reader to understand the indices and also help the writer to introduce the result and limits to this method. To catch the reader attention, it is necessary to present climate in Serbia earlier in the paper. For example, adding a map of Serbia who presents the location of the three stations studied and the cities presents in the paper. Information concerning the weather in Serbia should also be added: most severe droughts, theirs consequences on the populations... Results The result section is a numeration of the results. This section is the most important of the paper, it is important to bring more analysis to the presented results. Results of the analysis also need to be more accurate and explained. They also need to be more synthetic. For example, as the paper objective is to rate occurrence of thermal stress a trend of the UTCI indices could be computed. Correlation between the different indices obtained could be computed the complete the analysis. Presenting a figure showing the intensity of UTCI index values may help to improve this section and help to reach substantial conclusions.

Figures The ordinate of the figure 2, 3 and 4 need to be change to "number of days" instead of "Days". In general, figures and tables needs to be improved to make them

easier to understand.

Conclusion In the introduction of the paper, the role of such indexes is introduced (weather alerte, weather forcast). The conclusion should come back to the importance and role of this UTCI index values and the possible consequences of its increase on population.

Reference The following reference seems relevant and needs to be added: Bogdanović, D. and al. (2013). The impact of the July 2007 heat wave on daily mortality in Belgrade, Serbia. Central European journal of public health, 21(3), p. 140-145.

English language English language must be checked, many spelling mistakes are presents.

Line 9: Serbia Line 12: stations Line 16: a row, in July Line 17: in a row Line 21: in human body Line 22: biggest Line 30: human instead of man Line 34: wih the objective Line 35: thermal components are considered within UTCL Line 37: a heat budget index considering philological Line 41: what does stimuli means? Explanation are needed Line 58: "the idea of the analysis. . ..." this is relevant to your study? if yes, I will put if at the end of the section rather than lost there! Line 64: their connection with the circulation of the atmosphere, could you please give more information Line 67: "tropical days" give a definition Line 74: reference needs to be added Line 105: the second station is located in Nis. . . Line 107: Nisava valley located in the central part Line 107: and is situated Line 110: the third analysed station Line 116: multiday data Line 119: to compute UTCI Line 121: add a reference after "all spatial and temporal scales" Line 123: define "which was integrated with an adaptive clothing model" Line 128: 0.5 m.s-1 Line 130: 4 km.h-1, 135 W.m-2, explain 2.3MET Line 145: This section presents the results Line 157: during 20 years Line 158: certainly Line 195: 35°C Line 228: this is coherent with

---

## Referee Comment (RC2) · Anonymous Referee #2 · 14 Jan 2020

The assessment of bioclimatic conditions in Serbia has been investigated in the present research. This work seems to be interesting and can be accepted for publication after major revisions. The content of the manuscript should be deeply improved in terms of scientific (results, interpretations and discussions) and technical approaches (methodological approach). The content is deemed to be not fully satisfactory to understand the main finding of the work.

Introduction According to the introduction, many studies have been carried out in Serbia. It's important to underline the original contribution of the present research. This part should be improved and some part are difficult to understand Methodological approach Any detailed explanation has been presented in this part. I have not understood the methodology and how the results have been calculated. Then, a more recent me-

teorological data sets are used for the computation of the UTCI index values for the selected stations. Starting in 1998 should not be enough to calculate the long-term trends. Starting before should be more interesting to identify the trend of such extreme heat waves. Results This part should be more explained. I cannot fully understand the results since the methodology is not detailed. More analysis should be added to explain the main finding of the paper. Parts of Discussions and Conclusions Both parts should be written differently. The work should be discussed with other previous works dealing with extremes in this zone and also other works using the same methodology (if this one is not new.. I don't know exactly since it's not clearly presented).

---

## Author Comment (AC2) · 19 Feb 2020

Dear Reviewer, We would like to thank the you for giving us the opportunity to submit a revised draft of our manuscript titled Summer variation of the UTCI index and Heat Waves in Serbia to Journal of Natural Hazard and Earth System Science. We appreciate the time and effort that you dedicated to providing valuable feedback on our manuscript. We have been able to incorporate changes to reflect most of the suggestions. Since the you emphasized major revisions to the manuscript, significant changes have been made to the paper. The revision, based on your guidance, includes changed interpretation of research in all chapters.

Please also note the supplement to this comment:

[Figure]

https://www.nat-hazards-earth-syst-sci-discuss.net/nhess-2019-270/nhess-2019-270-AC2-supplement.pdf

---

## Author Response (AR1)

**10th April, 2020.**

Resubmission of manuscript "Summer variation of the UTCI index and Heat Waves in Serbia", **nhess-2019-270**

Dear Editor,

Thank you very much for your kind consideration of this resubmitted version of our manuscript. Please find enclosed our revised manuscript, entitled "Analysis of UTCI index during heat waves in Serbia" by Milica Pecelj, Milica Lukić, Dejan Filipović, Branko Protić, Uroš Bogdanović, for publication consideration in the Natural Hazards and Earth System Sciences (NHESS).

We value the comments received greatly, as they have pointed out a number of issues to be addressed. We would like to thank the editor and the reviewers for taking time in reading and suggesting modifications to the paper. We highly appreciate it, as the comments have been very useful to improve the paper.

We did significant modifications to the initial manuscript based on the suggestions of the reviewers. The modification has been done in all section in the manuscript:

- We have completely modified the Introduction, where we have emphasized and explained the concept of heat budget index more clearly. We have supplemented the introduction with a new text outlining the concept of heat budget indices and their importance for bioclimatic analysis. We modified the paragraphs followed by detailed recent studies (lines 47-58; lines 63-74; lines 77-94) and moved it into the Discussion section (lines 471-484; lines 438-454; lines 455-461).
- To provide a clearer introduction we have inserted a new subheading ''Study area''
- The section ''Materials and methods'' is improved with two subheadings about detailed explanation of UTCI index and indices considered in the study.
- The previous chapter ''Results and discussion'' are divided into two different ones, ''Results'' with inserted three subheadings and ''Discussion''.

It should be emphasized that in order to improve the initial manuscript, it has been included one more co-author whose contribution was related to the interpretation of the results and the improvement of the figures. To address the comments 5 and 6 from the reviewer RC1 regarding Results and Figures, we conducted more detail analysis between very strong heat stress category with tmax and interpreted all results with improved figures. This was executed by Uroš Bogdanović. Therefore, in the revised manuscript we have decided to include him as a co-author of the paper. This inclusion was discussed with and has received consent from all the co-authors of the paper.

For ease to follow the changes in revised manuscript, we uploaded another revised manuscript, where the modified text is colored green and the new added text is colored blue. Hopefully this will be useful for the new review.

We hope that the editor will find the paper suitable for publication.

Thank you again for consideration.

Yours sincerely,

Milica Pecelj (On behalf of the authors of the manuscript)

Please find enclosed

1. Response to the reviewer 1 (RC1)
2. Response to the reviewer 2 (RC2)
3. Marked up manuscript version

Response to peer reviewer comments
nhess-2019-270: **Summer variation of the UTCI index and Heat Waves in Serbia**
Milica M. Pecelj, Milica Z. Lukić, Dejan J. Filipović, Branko M. Protić

Dear Reviewer,

We would like to thank you for giving us the opportunity to submit a revised draft of our manuscript titled Summer variation of the UTCI index and Heat Waves in Serbia to Journal of Natural Hazard and Earth System Science. We appreciate the time and effort that you dedicated to providing valuable feedback on our manuscript.

We have been able to incorporate changes to reflect most of the suggestions. Since you emphasized major revisions to the manuscript, significant changes have been made to the paper. The revision, based on your guidance, includes changed interpretation of research in all chapters.

Here are the responses to yours comments and concerns, listed step by step. To make the changes in the manuscript easier to follow, we have highlighted the modified text in green and the new added text in blue.

**Comment 1: Title**

*To me the title could be: "Analysis of UTCI index during heat waves in Serbia"*

**Author response**: Thank you. We agree with this suggestion and have made changes, so the revised title reads exactly as suggested: Analysis of UTCI index during heat waves in Serbia

**Comment 2: General comment**

*This work could potential be interesting however improvement of the paper is need previous to publication in order to reach the expected international standards requested by the journal. First of all, according to the elements presented in the introduction, many similar studies have already been conducted in Serbia. Within the current study, a more recent meteorological data sets are used to compute the UTCI index values for the selected stations. Starting before 1998 would have been even more interesting to identify the trend of such extreme heat waves.*

**Author response**: Thank you for pointing this out. We agree with the reviewer's statement however, in the case of our study, it seems slightly out of scope because the procedure for obtaining daily hourly data from the national weather center is very slow, so it would be impossible to respect the time limit for correction. It would certainly be interesting to explore this aspect for future research. Since many studies have been conducted in Serbia concerning the analysis of heat waves in the last 50 years, we considered their results relevant as they indicate an increase in heat waves, especially since 2000. Starting from this fact, we used the data available only from 1999 to 2017 and for this period we did a bioclimatic identification of thermal discomfort. We changed a part in introduction to emphasize this point.

The revised text reads as follows on:

(lines 77-81) Considering extreme temperature is one of the most significant climatic parameters in the universal context of climate change, analyses of heat waves in Serbia have been performed in several different approaches and all of them show a growing trend in max temperature (Drljača et al., 2009; Unkašević and Tošić, 2009a; Unkašević and Tošić, 2009b; Unkašević and Tošić, 2013; Malinović-Milićević et al., 2013) and heat waves (Unkašević and Tošić, 2011; Unkašević and Tošić, 2015 Milićević-Malinović et al., 2016) especially since 2000.

Keeping this fact in mind, the basic idea of this study is to conduct the analysis of human bioclimatic conditions in Serbia in the last 20 years, where according to the previous studies mentioned above, warming has been perceived and recorded, especially since 2000. Determining extreme weather event from the aspect of heat budget indices

allows other meteorological parameters to be taken into account because high temperatures and humidity generate a heat load more rapidly, unless the wind and direct radiation are taken into account.

(lines 89-93) The conducted human bioclimatic evaluation of UTCI thermal stress in Serbia was considered to be of great importance due to the identification of biothermal heat hazard and the study of the evolution of such indices regarding climate change. This evaluation is aimed at addressing the following topics in this study:

- providing a comprehensive assessment of the human heat stress associated with UTCI index and
- comparing it to heat waves defined by meteorological parameters.

**Comment 3: Abstract/Introduction**

*The abstract is concise and seems complete. However, the introduction needs to be improved. The first part introduces the UTCI index, then heat waves in Serbia are presented, followed by recent studies (line 75 to 94). This final part of the introduction is difficult to understand. It is better to rewrite it and introduce that there is a need to study the evolution of such indices regarding climate change, as trend has been identified during the period 1999-2007.*

**Author response**: Thank you for pointing this out. We agree with this comment. Therefore, in relation to your remarks, we have completely modified the introduction, where we have emphasized and explained the concept of heat budget index more clearly. We have supplemented the introduction with a new text outlining the concept of heat budget indices and their importance for bioclimatic analysis (lines 35-52).

The revised text reads as follows on:

(lines 32-38) Over the last century many models and indices have been developed for the assessment of human exposure to heat, ranging from simple physical instruments designed to imitate the human heat exchange with the environment, to complex thermophysiological models that simulate external and internal body heat transfer and allow detailed simulation of different work load, clothing, and climate scenarios (Havenith and Fiala, 2015).

(lines 39-52) Many thermal indices have been developed for the purpose of describing the complex conditions of heat exchange between the human body and its thermal environment (Fanger, 1970; Landsberg, 1972; Parsons, 2003). Those are two meteorological parameter indices – the ones used for cold environment conditions combining the air temperature and the wind velocity (Osczevski and Bluestein, 2005; Siple and Passel, 1945;), and the ones used for heat environment conditions combining the air temperature and humidity (Masterton and Richardson, 1979; Steadman, 1984; Yaglou and Minard, 1957). Considering their shortcomings, i.e. the fact that they are not universally applicable to all climatic regions, including different seasons, the dominance in the analysis of biothermal conditions and thermal stress over the last 30 years has been taken over by so-called heat budget indices (Blazejczyk, 1994; Hoppe, 1999; Jendritzky, 2012). The heat budget indices are based on the thermal exchange between humans and the environment. Methodologically, they use variables related to meteorological (air temperature, wind velocity, radiation, air humidity) and physiological processes (most commonly metabolic heat) and clothing insulation. The issue that should be emphasized during the analysis of the thermal stress is the influence that extreme temperatures have on the physiological parameters in humans. According to McGregor and Vanos (2018), generated heat load can undermine the human body's ability to hold its core temperature within the range of optimal physiological achievement.

Then, we modified the paragraph followed by detailed recent studies (lines 47-58; lines 63-74; lines 77-94) and moved it into the Discussion section (lines 471-484; lines 438-454; lines 455-461).

More specifically, lines 47-58 have been moved into the discussion lines 471-484, where a reference about heat wave related mortality in 2007 by Bogdanović et al. (2013) was added (lines 485-498).

The revised text reads as follows on:

(lines 471-484) Identified biothermal discomfort in July and August as the hottest summer months in Serbia, could provoke the health disorders, more frequent. Specific characteristics of HWs such as intensity and duration has been

found to have devastating impacts on human health and wellbeing. In certain cases, heat waves cause problems, especially to children, elderly people, chronic patients and workers. Analysis of the UTCI index correlated with mortality in Europe shows the deaths occur in the southern Europe (Italy, Portugal, Greece, Spain) for the categories of moderate and strong stress. There is an impact of heat load in increasing mortality, as conditions become more thermally stressful (Di Napoli et al., 2018). The intensity of heat stress in Poland was assessed by the UTCI index based on daily mortality and weather data for 1991–2000. There is increase in mortality for the days with strong and very strong heat stress, in relation to no thermal stress days, of 12% and 47%, respectively (Błażejczyk et al., 2017). Biothermal conditions and mortality rates show that during summer in southern Europe thresholds of Physiological Subjective Temperature (PST) increase when moving from west to east. The general founding is that population of central Europe is more sensitive for thermal stimuli in summer then citizens of the south of continent (Błażejczyk and McGregor, 2008). The statement about the PST index is in support with the comparison of UTCI index with other thermal indices, including PST index, which is very well correlated with UTCI index (Błażejczyk et al., 2012).

(lines 485-498) Similarly, Bogdanović et al. (2013) reported a significant short-term excess mortality on 16th July in Belgrade, when the maximum daily temperature exceeded 35°C, leading to 167 excess deaths (38% increase compared to the number of expected deaths) for nine consecutive days of heat, with a progressive return to almost normal mortality as the maximum temperature dropped below 35°C on 25th July.

In addition, lines 63-74 have been moved into the discussion lines 438-454, and there is an explanation of what a tropical day is (line 456-457) in the analysis by Unkasevic and Tosic (2009b).

The revised text reads as follows on:

(lines 438-454) This agrees with findings of Unkašević and Tošić (2011, 2015) that trend of heat waves in Serbia is increase, especially since 2000. In order to justify present research for the last 20 years, the earlier results related to the increase in heat waves will be discussed in more detail. In that regard, previous research of heat waves in Niš for 1949-2007, based on the autoregressive-moving-average model, show observed the warmest years during three periods, in 1951-1952, 1987-1998 and 2000-2007 (Unkašević and Tošić, 2009b). According to this research, the longest heat wave was recorded in 1952 in Niš with 21 days while during 2003 in was observed 29 consecutive tropical days. It should be considered that according to the Serbian National Weather Service, tropical days are days with a maximum temperature over 30°C. Further, based on Heat Wave Duration index (HWD) i.e. daily maximum values of air temperature, Drljača et al. (2009) determined the length and strength of heat waves in Niš during the summer season. The analysis showed Niš with the greater number of heat waves compared to Belgrade larger urban area. As stated in research, since the mid-1980s, heat waves have a higher frequency and occur on average every year. Prior to that period heat wave fluctuations have generally been reported in one of two years (Niš during summer) (Drljača et al., 2009). An increase of the heat wave duration, in addition to increase of the heat wave frequency of occurrence was detected by Unkašević and Tošić (2011) during the period 1999 – 2007, studying the characteristics of the heat waves in central Serbia (1949 – 2007).

Lastly, lines 77-94 have been moved into the discussion section (lines 455-461). The revised text reads as follows:

(lines 455-461) As regards other heat budget index of Heat Load (HL) in Serbia (July) for 2000-2010 is noted that in, Niš, Zlatibor and Novi Sad expired "extremely hot" in 2007 and 2000 (Milovanović et al., 2017). Referring to heat budget index Physiological Equivalent Temperature PET for the period 1949–2012, the highest number of heat waves in Novi Sad was observed in the last two decades and in the first decade of the investigated period (Basarin et al., 2016) while the number of the days above particular thresholds for 1961-2014, show increase along with a number of heat waves per year since 1981. The highest values of mean annual PET are mostly detected during the last 15 years. (Basarin et al., 2018).

To provide a clearer introduction we have inserted a new text. Namely, we added a heading ''Study area" where we provided a relief map of Serbia, inside Europe, with the location of the three stations studied in the paper (lines 95-105). Basic information about the climate of Serbia has been added. Thus, we switched the first paragraph from the Materials and methods section (lines 100-115), which in the revised version takes up lines 106-120.

The revised text reads as follows:

(lines 95-105) Study area

The Republic of Serbia is located in south-eastern Europe, in the area of the southern Pannonian Plain and the central Balkans. Northern Serbia is mainly flat, while its central and southern areas consist of highlands and mountains (Gocić and Trajković, 2014), as the map of the relief characteristics shows (Figure 1). The territory of Serbia is characterised by temperate continental and mountain climate and the spatial distribution of climate parameters is determined by geographical location. The research involved studying three synoptic stations located in different geographical areas of Serbia: (i) Novi Sad, (ii) Niš and (iii) Mt. Zlatibor (Figure 1).

(lines 106-120) The first weather station (at an altitude of 86 m) is located in the territory of Novi Sad, which is the administrative urban center of Vojvodina province and South Bačka District (Figure $1_1$). The city is located in the southern part of the Pannonian Basin, on the Danube River bank near Mt. Fruška Gora, and the national park bearing the same name. Novi Sad has a temperate continental climate, summers are warm and winters are cold with a small amount of snow (Lazić et al., 2006). The second station is Niš (at an altitude of 202 m) and it is located in the Niš Fortress. This city is the administrative urban center of the Nišava District in southern Serbia and it is situated in the Nišava valley, located in the central part of the spacious geological depression called the Nišava Basin (Figure $1_2$). According to Köppen's climate classification, the Nišava valley belongs to the Cfwax type – the Danube type of moderately warm and humid climate characterized by hot summers (the highest precipitation is recorded at the beginning of the summer) and somewhat dry winters (Prokić, 2018). Compared to other cities in this valley (Dimitrovgrad, Pirot and Bela Palanka), Niš is the hottest one with an average annual temperature of 11.8°C (Prokić, 2018). The third station is Zlatibor, a mountain in western Serbia, which belongs to the mountain range of the Dinaric Alps (Figure $1_3$). Zlatibor weather station lies at an altitude of 1029 m a.s.l. In the area of Zlatibor plateau mountains meet air currents from the Adriatic Sea, which, as it can be assumed, creates a favorable climate and for this reason Zlatibor has already been confirmed as a climatic resort (Pecelj et al., 2017). The mountain climate of Zlatibor involves long and cold winters, short and fresh summers, and less pronounced spring and autumn.

[Figure]

**Figure 1**. Relief map of Serbia with the areas studied: 1-Novi Sad, 2-Niš, 3-Zlatibor and a map of geographical location of Serbia in Europe

Maps were created using QGIS 3.8 software on the basis of the official European Commission's (Eurostat) datasets, available at https://ec.europa.eu/eurostat/web/gisco/geodata/reference-data/, (Map ratio: 1:1900000; Map projection: WGS 84/UTM, Zone 34N, the official national coordinate system)

**Comment 4: Materials and methods**

*Materials and methods: The model used to compute the index is not explained and presented. The reader does not know how the index are computed and therefore cannot judge the methodology and evaluate the results. Therefore, within the material and method section, it is needed to add the equations of the model. Moreover, to me a validation of the model should be introduced in the section, this will give more weight to the results, moreover this will help the reader to understand the indices and also help the writer to introduce the result and limits to this method. To catch the reader attention, it is necessary to present climate in Serbia earlier in the paper. For example, adding a map of Serbia who presents the location of the three stations studied and the cities presents in the paper. Information concerning the weather in Serbia should also be added: most severe droughts, their consequences on the populations...*

**Author response**: We have, accordingly, a modified description of the model used to compute the index and we have made changes, clarifying the index analysis methodology. More specifically, we have divided the Materials and methods into two sections. In the beginning, an overview of meteorological and physiological components is given (lines 122-131), where we describe in more detail the metabolic rate as a key physiological parameter. The revised text reads as follows:

(lines 122-131) The present study implements the methodological approach of Universal Thermal Climate Index (UTCI) based on human heat budget model relying on the evaluation of human energy balance. As a thermal comfort indicator, the UTCI considers combined meteorological and physiological parameters describing thermal comfort through the evaluation of human energy balance. In terms of physiological conditions, the metabolic rate plays the most important role. Metabolic processes in the human body create heat that is constantly exchanged with the environment achieving a state of thermal equilibrium in the body for maintaining a constant body temperature. The amount of the heat that is produced and released depends on the following: physical activity, clothing, sex, age, body mass, diet, mental state, health, external conditions, acclimatization, etc. As a measure of physical activity, a unit of "MET" was defined, which corresponds to the release of human heat of 58.2 $W.m^{-2}$ for average body surface area (1.8 $m^2$), i.e. it is equal to the energy rate produced per unit surface area of an average person seated at rest (ANSI/ASHRAE Standard 55). According to ISO 8996 for standard applications, the metabolic heat energy is M = 135 $W.m^{-2}$ i.e. 2.3 MET, for a person moving at a speed of $1.1 m.s^{-1}$.

The first section deals with the Universal Thermal Climate Index (UTCI), where we provide detailed definition, parameter inputs, and their limitations including stress classification (lines 135-142; lines 144-165). The revised text reads as follows:

(lines 133-140) The Universal Thermal Climate Index (UTCI)

For valid assessment of the outdoor thermal environment in the fields of public weather services, public health systems, urban planning, tourism and recreation and climate impact research, the Universal Thermal Climate Index (UTCI) has been developed based upon the most recent scientific progress in human thermo-physiology in biophysics and heat exchange theory. The UTCI is the result of an approach which was developed in the International Society of Biometeorology (ISB) Commission 6 and was later improved by COST Action 730 (Jendritzky et al., 2012). The design of UTCI is of great importance due to the fact that it is applicable to all seasons and climates together with all spatial and temporal scales.

(lines 144-165) In other words, this model simulates the same sweat production or skin wettedness in human body response as the actual environment condition (Błażejczyk et al., 2013; Błażejczyk et al., 2014). This is derived from the multi-dimensional dynamic response of a state-of-the-art multi-node thermo-physiological model of human heat transfer and thermoregulation (Fiala et al., 2012). The UTCI can be represented in general function as bellow:

UTCI = f (t, f, v, $t_{mrt}$) (Eq.1)

[revised manuscript text omitted]

**Comment 5: Results**

*The result section is a numeration of the results. This section is the most important of the paper, it is important to bring more analysis to the presented results. Results of the analysis also need to be more accurate and explained. They also need to be more synthetic. For example, as the paper objective is to rate occurrence of thermal stress a trend of the UTCI indices could be computed. Correlation between the different indices obtained could be computed the complete the analysis. Presenting a figure showing the intensity of UTCI index values may help to improve this section and help to reach substantial conclusions.*

**Author response**: We have divided the section Results and discussion in two separate sections. According to the 3 stages of the study, described in revised Methodology (Data and indices considered in the study), the results are presented in three sections: (i) UTCI indices ($UTCI_{avg}$, $UTCI_{7h}$, $UTCI_{14h}$), (ii) UTCI heat stress indices (strong heat stress SHS, very strong heat stress VSHS, extreme heat stress EHS) and thermal indices (hot day HD, hot night HN), (ii) Heat waves (HWE and VSHSE).

In all three sections the results are explained in more detail along with the modified and new added figures. Alongside the trend of UTCI index, the correlation between UTCI and maximum air temperature, is provided, too.

The revised text reads as follows:

[revised manuscript text omitted]
 Niš | July Novi Sad | Aug Niš | Aug Novi Sad | Sep Niš | VSHSE | July Niš | July Novi Sad | Aug Niš | Aug Novi Sad | Sep Niš |
|---|---|---|---|---|---|---|---|---|---|---|---|
| 1998 | 4 | / | 5 | / | / | 1998 | 7 | / | 5 | / | / |
| 1999 | / | / | 3, 4 | / | / | 1999 | / | / | 5 | / | / |
| 2000 | 3 | / | 4, 6 | 6 | / | 2000 | / | / | 6 | 7 | / |
| 2001 | 3 | / | 3, 3 | / | / | 2001 | / | / | / | / | / |
| 2002 | 6 | / | / | / | / | 2002 | / | / | / | / | / |
| 2003 | 4 | / | / | / | / | 2003 | / | / | / | / | / |
| 2004 | 3, 4 | 3 | / | / | / | 2004 | 5,5 | / | / | / | / |
| 2005 | 6 | / | / | / | / | 2005 | 6 | / | / | / | / |
| 2006 | / | / | 3 | / | / | 2006 | / | / | / | / | / |
| 2007 | 3, 10,4 | 6 | 6 | / | / | 2007 | 10 | 7 | / | / | / |
| 2008 | / | / | 4 | 3 | 3 | 2008 | / | / | 5 | / | / |
| 2009 | / | / | 3 | / | / | 2009 | / | / | / | / | / |
| 2010 | / | / | 4 | / | / | 2010 | / | / | 5 | / | / |
| 2011 | 3 | 4 | 3 | / | / | 2011 | / | / | / | / | / |
| 2012 | 9 | 6 | 4, 6 | 3, 5 | / | 2012 | 12 | 6 | 5,6 | 5,5 | / |
| 2013 | / | / | 7 | 4 | / | 2013 | / | / | 8 | 8 | / |
| 2014 | / | / | / | / | / | 2014 | / | / | / | / | / |
| 2015 | 7 | 9 | 6, 4 | 4, 5 | 5, 3 | 2015 | 10 | 8 | 7 | 6 | 5 |
| 2016 | 3 | / | / | / | / | 2016 | / | / | / | / | / |
| 2017 | 4, 4 | / | 6, 4 | 6, 3, 3 | / | 2017 | / | / | 6,5 | 6 | / |

Note: pink cells have more than one HWE and VSHSE duration defined by numbers of the consecutive days

Of the total number of hot days, it amounts to 21 HDs in September in Niš, only one HD was recorded in the first decade (1998-2007) of the investigated period, while the remaining HDs were recorded over the last decade (2008-2017), especially in 2011 and 2015. In the second decade of the investigated period the year 2015 stands out with two HWEs (September 1st to 5th, and 17th to 19th). In the same year the highest daily temperature was recorded in September (on September 18th, $t_{max}$=37.5°C).

(lines 369-384) Bioclimatic conditions analyzed by means of the UTCI index show that the calculated data observed at 14 CET are related to the marked heat wave events (HWE). When UTCI$_{14h}$ thermal stress category of "very strong heat stress" (VSHS) is compared to the selected heat wave events (HWE,) a sub-index is defined. It is called very strong heat stress event (VSHSE) and it is caused by the occurrence of 5 consecutive VSHS days. The VSHSE corresponds to heat wave event and provokes a severe biothermal discomfort, so it was used as an indicator of extremely unfavorable bioclimatic conditions (biothermal heat hazard). Along these lines, the VSHSE in Niš occurred during July in 1998, 2004, 2005, 2007, 2012 and 2015 where 7, 5 (on two occasions), 6, 10, 12 and 10 VSHS days in a row were recorded respectively (Table 4, VSHSE). In August, VSHSE was recorded on 5 VSHS days in a row in 1998, 1999, 2008, 2010, 2012 and 2017, while in 2000, 2012, 2013, 2015 and 2017 6, 6, 8, 7 and 6 VSHS days in a row were recorded respectively. It should be emphasized that there were two VSHSE events in 2012 and 2017, for each year 5 and 6 VSHS days in a row. September 2015 saw a VSHSE with 5 VSHS days in a row. According to Unkašević and Tošić (2009b) the highest temperatures in Niš ever were recorded during the summer of 2007 (44.2°C) and the summer of 2000 (42.5°C), covering the data period from 1948 to 2007. The VSHSE occurred in Novi Sad in July 2007 (7 VSHS days in a row), 2012 (6 VSHS days in a row) and 2015 (8 VSHS days in a row) and in August 2000 (7 VSHS days in a row), 2013 (8 VSHS days in a row), 2015 (6 VSHS days in a row) and 2017 (6 VSHS days in a row). In August 2012 VSHSE occurred twice and each time 5 VSHS days in a row were recorded. There was no sub-index VSHSE recorded in Zlatibor.

**Comment 6: Figures**

*The ordinate of the figure 2, 3 and 4 need to be change to "number of days" instead of "Days". In general, figures and tables needs to be improved to make them easier to understand.*

**Author response**: We agree with this suggestion and we have improved the figures and tables. Figures 2,3 and 4 have been changed and merged to form one, which became Figure 4 in the revised manuscript. Furthermore, one new figure (Figure 3 in the revised version) has been added in the first section of Results to represent the trend of the UTCI$_{14h}$ index. Table 3 has been supplemented with VSHSE index compared with HWE. In the revised manuscript it became Table 4. The revised figures and tables are provided in the previous author response (response for Comment 5: Results).

**Additional clarifications**

Considering that the revised version has separated the results and the discussions, we need to point out the changes in the discussion section. The revised text is provided as follows:

[revised manuscript text omitted]

**Comment 7: Conclusion**

*In the introduction of the paper, the role of such indexes is introduced (weather alert, weather forecast). The conclusion should come back to the importance and role of this UTCI index values and the possible consequences of its increase on population.*

**Author response**: We have incorporated your suggestion throughout the conclusion section. The revised text is provided as follows:

(Lines 497-510) The present research of the human bioclimatic evaluation of UTCI thermal stress in Serbia is considered to be of great importance due to the identification of biothermal heat hazard and the study of the evolution of such indices regarding climate change. This evaluation aims at providing a comprehensive assessment of the human heat stress associated with UTCI index and heat waves defined by UTCI very strong heat stress and maximum air temperature.

The assessment of human biothermal conditions for the investigated period from 1998 to 2017 was provided for three synoptic stations – two urban (Niš and Novi Sad) and one station representing mountain areas of an altitude up to 1500 m (Zlatibor). All of them are located in different geographical areas in Serbia. The results obtained during the study indicate the significant increase in biothermal discomfort associated with heat waves defined by UTCI "very strong heat stress". It was found that the most frequent heat stress categories are "strong heat stress" (SHS) and "very strong heat stress" (VSHS) and they have occurred in all three locations with the tendency of increase in the number of days in the last decade. The VSHS describes an alarming biothermal state and has occurred frequently in the last ten years, particularly in Niš and Novi Sad. The outcome of the study is $UTCI_{14h}$ sub-index "very strong heat stress event" (VSHSE) as an indicator of biothermal heat hazard. Undoubtedly, heat waves (HWs) are one of the natural hazards with increasing impact in urban areas related to higher population density.

(lines 514-519) Considering these facts, it can be deduced that heat waves are becoming more frequent, stronger and longer. Thus, frequent heat waves since 2007 certainly indicate biothermal heat hazard. The more lasting events of bioclimatic discomfort will indicate more stressful bioclimatic conditions for human health and wellbeing. However, the results of the study highlight the importance of UTCI index as a bioclimatic indicator of biothermal hazard in Serbia particularly if it can serve as one of the criteria for initiating heat warnings in Serbia.

**Comment 8: Reference**

*Reference: The following reference seems relevant and needs to be added: Bogdanović, D. and al. (2013). The impact of the July 2007 heat wave on daily mortality in Belgrade, Serbia. Central European journal of public health, 21(3), p. 140-145.*

**Author response**: We have added the suggested reference in the Discussion section (lines 492-495). The revised text is provided as follows:

[revised manuscript text omitted]

**Comment 9: English**

*English language must be checked, many spelling mistakes are presents.*

**Author response**: With regard to the above comment, all the spelling and grammatical errors pointed out by the reviewers have been corrected.

Response to peer reviewer comments
nhess-2019-270: **Summer variation of the UTCI index and Heat Waves in Serbia**
Milica M. Pecelj, Milica Z. Lukić, Dejan J. Filipović, Branko M. Protić

Dear Reviewer,

We would like to thank you for giving us the opportunity to submit a revised draft of our manuscript titled Summer variation of the UTCI index and Heat Waves in Serbia to Journal of Natural Hazard and Earth System Science. We appreciate the time and effort that you dedicated to providing valuable feedback on our manuscript.

We have been able to incorporate changes to reflect most of the suggestions. Since you emphasized major revisions to the manuscript, significant changes have been made to the paper. The revision, based on your input, includes changed interpretation of research in all chapters.

Here are the responses to yours comments and concerns, listed step by step. To make the changes in the manuscript easier to follow, we have highlighted the modified text in green and the new added text in blue.

**Reviewer Comment:**

*The assessment of bioclimatic conditions in Serbia has been investigated in the present research. This work seems to be interesting and can be accepted for publication after major revisions. The content of the manuscript should be deeply improved in terms of scientific (results, interpretations and discussions) and technical approaches (methodological approach). The content is deemed to be not fully satisfactory to understand the main finding of the work. Introduction According to the introduction, many studies have been carried out in Serbia. It's important to underline the original contribution of the present research. This part should be improved and some part are difficult to understand. Methodological approach Any detailed explanation has been presented in this part. I have not understood the methodology and how the results have been calculated. Then, a more recent meteorological data sets are used for the computation of the UTCI index values for the selected stations. Starting in 1998 should not be enough to calculate the long-term trends. Starting before should be more interesting to identify the trend of such extreme heat waves. Results This part should be more explained. I cannot fully understand the results since the methodology is not detailed. More analysis should be added to explain the main finding of the paper. Parts of Discussions and Conclusions Both parts should be written differently. The work should be discussed with other previous works dealing with extremes in this zone and also other works using the same methodology (if this one is not new.. I don't know exactly since it's not clearly presented).*

**Author response:** Thank you for pointing this out. Since the major revisions to the manuscript have emphasized, significant changes were made to the manuscript. The revision, based on the review input, includes a number of following changes:

**Title**

We have modified the title and named in revised text as: Analysis of UTCI index during heat waves in Serbia

*Abstract*

*According to all changes that have made in the manuscript we have modified Abstract* and in revised text is provided as follows:

(lines 8-9) The objective of this paper is to assess the bioclimatic conditions in Serbia so as to identify biothermal heat hazard.

(lines 12-14) In order to identify patterns of biothermal heat stress conditions, the thresholds of daily UTCI index, refers as very strong heat stress (VSHS), are compared with the thresholds of daily maximum of air temperature which are termed as heat wave events.

(lines 16-18) Heat wave events recorded (HWE, VSHSE) in July 2007 (10 days), 2012 (9, 12 days) 2015 (7, 10 days) were of the longest duration and refer as indicators of biothermal heat hazard.

**Introduction**

In relation to your remarks, we have completely modified the introduction where have emphasized and explained the concept of heat budget index more clearly. We have supplemented the introduction with a new text outlining the concept of heat budget indices, their importance for bioclimatic analysis.

We agree with the reviewer statement about data set used in the study however, in the case of our study, it seems slightly out of scope because the procedure for obtaining daily hourly data from the national weather center is very slow, so it would be impossible to respect the time limit for correction. It would certainly have been interesting to explore this aspect for future research. Since many studies have been conducted in Serbia concerning the analysis of heat waves in the last 50 years, we considered their results relevant indicating an increase in heat waves especially since 2000. Starting from this fact, we used available data only from 1998-2017 and for that period we did a bioclimatic identification of thermal discomfort. We changed a part in introduction to emphasize this point.

Then, we modified the paragraph followed by detailed recent studies (lines 47-58; lines 63-74; lines 77-94) and moved it into the discussion section (lines 479-494; lines 448-462; lines 463-469). More specifically, lines 47-58 have been moved into the discussion lines 479-494, where a reference about heat wave related mortality in 2007 by Bogdanović et al. (2013) was added (lines 491-494). In addition, lines 63-74 have been moved into the discussion lines 448-462, and there is an explanation of what a tropical day is (line 455-456) in the analysis by Unkasevic and Tosic (2009b). Lastly, lines 77-94 have been moved into the discussion section (lines 466-472).

The revised text reads as follows on:

[revised manuscript text omitted]

**Results**

We have divided the section Results and discussion in two separate sections. According to the 3 stages of the study, described in revised Methodology (Data and indices considered in the study), the results are presented in three sections: (i) UTCI indices ($UTCI_{avg}$, $UTCI_{7h}$, $UTCI_{14h}$), (ii) UTCI heat stress indices (strong heat stress SHS, very strong heat stress VSHS, extreme heat stress EHS) and thermal indices (hot day HD, hot night HN), (ii) Heat waves (HWE

and VSHSE). In all three sections the results are explained in more detail along with the modified and new added figures. Alongside the trend of UTCI index, the correlation between UTCI and maximum air temperature, is provided, too.

We have improved the figures and tables. Figures 2,3 and 4 have been changed and merged to form one, which became Figure 4 in the revised manuscript. Furthermore, one new figure (Figure 3 in the revised version) has been added in the first section of Results to represent the trend of the UTCI$_{14h}$ index. Table 3 has been supplemented with VSHSE index compared with HWE. In the revised manuscript it became Table 4.

The revised text reads as follows:

[revised manuscript text omitted]

Considering that the revised version has separated the results and the discussions, we need to point out the changes in the discussion section. According to the mention changes regarding Introduction chapter, we modified the paragraph followed by detailed recent studies (lines 47-58; lines 63-74; lines 77-94) and moved it into the discussion section (lines 479-494; lines 448-462; lines 463-469). More specifically, lines 47-58 have been moved into the discussion lines 479-494. In addition, lines 63-74 have been moved into the discussion lines 448-462, and there is an explanation of what a tropical day is (line 456-457) in the analysis by Unkasevic and Tosic (2009b). Lastly, lines 77-94 have been moved into the discussion section (lines 463-469).

The revised text is provided as follows:

**Discussion**

[revised manuscript text omitted]

**Reference**

Considering that the revised version has significant changes the supplemented reference in the revised version is added as follows:

[revised manuscript text omitted]

---

## Editor Decision (ED1)

**Summer variation of the UTCI index and Heat Waves in Serbia**

Dear Authors

I appreciate the efforts realized to improve the manuscript. Parts of the paper have been substantially improved. However, the clarifications brought still minor and need to be deeply explained to make easier the understanding of this research for the readers. May main concern is the possible acceptance of the manuscript after a major revision.

The work focuses locally to Siberia. Which extension? The extend of this work for other regions?
My main global comments at this stage are:

- **Materials and methods**: a figure illustrating the data and their statistal behaviour in required.
A map showing the location of the study area and the extension of the climate patterns is strongly required.
- **Results and discussions:** deep clarifications are required to point the limitations of the method and the different approximations used.

At this stage, the discussion and conclusion are still very confusing and not fully supported by the results.
My feeling is that you want to say too many things (from some results). You ought to focus on one or two main ideas and these ideas should to be supported by the results.
You are going away in the interpretation. May be other analyses are required to largely support the discussions.

The section of results and Discussions might benefit from substantial extension; the content is very limited to be considered as a discussion of an original research.

- English is not my main language. Although improvements seem to have been made, it still seems to me that further improvements could be made. I strongly suggest a thorough review of the text by other native English speakers.

---

## Author Response (AR2)

**19ᵗʰ May, 2020.**

Resubmission of manuscript "Summer variation of the UTCI index and Heat Waves in Serbia", **nhess-2019-270**

Dear Editor,

Thank you very much for your kind consideration of this resubmitted version of our manuscript. Please find enclosed our revised manuscript, entitled "Analysis of UTCI index during heat waves in Serbia" by Milica Pecelj, Milica Lukić, Dejan Filipović, Branko Protić, Uroš Bogdanović, for publication consideration in the Natural Hazards and Earth System Sciences (NHESS).

We highly appreciate your suggestions. We did minor modifications to the manuscript based on the suggestions. The modification on the manuscript has been done by following:

1. First, we hope that you made technical error when typed the name of the country. Instead of Serbia stand Siberia in the text. Our paper focuses to Serbia in Balkan peninsula (Southeastern Europe).

2. Regarding map showing location of the study area, in the previous revised version we have added a map and named it as Figure 1 in the Introduction section (Subheading-Study area). The map shows geographical location of the study area.

    However, in this revised version, we supplemented Figure 1. with another map showing an average number of tropical days in Serbia.  In introduction section we added small paragraph (lines 28-32), a sentence (lines 105-107) and supplemented the legend of Figure 1. (lines 111-112)

    The revised text reads as follows on:

[revised manuscript text omitted]